

# Long-term visibility variation in Athens (1931-2013): A proxy for local and regional atmospheric aerosol loads

Dimitra Founda[1], Stelios Kazadzis[1,2], Nikolaos Mihalopoulos[1,3], Evangelos Gerasopoulos[1], Maria Lianou[1]

[1]Institute for Environmental Research & Sustainable Development, National Observatory of Athens, Greece
[2]Physikalisch-Meteorologisches Observatorium Davos, World Radiation Center, Switzerland
[3]Department of Chemistry, University of Crete, Greece

*Correspondence to*: Dimitra Founda (founda@noa.gr)

**Abstract.** This study explores the inter-decadal variability and trends of surface horizontal visibility at the urban area of Athens from 1931 to 2013, using the historical archives of the National Observatory of Athens (NOA). A prominent deterioration of visibility in the city was detected, with the long- term linear trend amounting to -2.8 km decade$^{-1}$ (p < 0.001), over the entire studied period. This was not accompanied with any significant trend in relative humidity (RH) or precipitation over the same period. A slight recovery of visibility levels seems to be established in the recent decade (2004-2013). It was found that very good visibility (> 20 km) occurred at a frequency of 34 % before the 1950s, while this percentage drops to just 2 % during the recent decade. The rapid impairment of the visual air quality in Athens around 1950, points to the increased levels of air pollution from local and/or regional emission sources, related to high urbanization rates and/or higher rates of anthropogenic emissions increase on a global scale at that period. A marked seasonal cycle was detected in visibility before the1950s, which attenuates afterwards. Visibility was found to be negatively/positively correlated with relative humidity (RH)/wind speed, the correlation being statistically valid at certain periods. Wind regime and mainly wind direction and corresponding air masses origin was found to highly control visibility levels in Athens. The comparison between visibility in Athens and at a reference, non urban site, revealed similar negative trends over the common period of observations, suggesting that apart from the contribution of local sources, visibility in Athens is highly determined by aerosol loads of regional origin. Satellite derived aerosol optical depth (AOD) retrievals over Athens since 2000, and surface measurements of PM$_{10}$ confirmed the relation of visibility with aerosol loads.





## 1 Introduction

Visibility is defined as the greatest distance at which a black object of suitable dimensions (located on the ground) can be seen and recognized, when observed against the horizon sky during daylight, (WMO, 1992). Visibility represents one of the dominant features of the climate and landscape of an area. Although it is highly affected by atmospheric circulation and the prevailing meteorological conditions, under clear sky conditions it is mainly determined from the loading of atmospheric aerosols (Davis, 1991; Lee, 1994; van Beelen and van Delden, 2012; Doyle and Dorling, 2002; Singh and Dey, 2012), therefore, visibility can be a strong indicator of air quality at an area. Horizontal visibility has been also introduced in formulas for the estimation of atmospheric turbidity parameters (e.g. in the Ångström atmospheric turbidity coefficients, Eltbaakh et al., 2012).

Aerosols in the atmosphere contribute to light extinction by scattering and absorbing, thus they reduce visibility (Appel et al., 1985; Chan et al., 1999; Elias et al., 2009; Singh and Dey, 2012). The impact of particulate matter on visibility depends on its physical (e.g. particle size distribution) and chemical properties (Dayan and Levy, 2005). In particular, visibility is inversely related to light extinction coefficient, which is determined from scattering and absorption of light by gases and particles, the latter (e.g. sulphate and carbon containing particles) being the main contributor (Malm, 1999; Hand et al., 2002; Baumer et al., 2008; Deng et al., 2011; Wang et al., 2012). Sulphate and carbon containig particles have a major role in light absorption, while the role of relative humidity (RH) on visibility is also important (Larson and Cass, 1989; Malm, 1999), as when RH reaches saturation values, visibility deteriorates due to fog formation and the hygroscopic growth of $SO_4^{2-}$, $NH_4^+$ and $NO_3^-$ particles (Tang, 1996; Sing and Dey, 2012). At the local to regional level, wind speed and direction are also very important factors, as they determine the transport and origin of air pollution.

Although the use of visibility as a viable atmospheric variable has been disputed by many researchers due to the numerous biases related to observational procedures (Davis, 1991), visibility statistics have been increasingly used as a surrogate for aerosol loads (Zhao et al., 2011), especially since visibility records span quite long-term periods. Today, there is a large number of studies that use visibility observations to investigate the spatial and temporal variation of the optical properties of the atmosphere, mainly in relation to pollutants emissions and aerosol loads. Studies refer to global, regional and local scales. On a global scale, a decrease of clear sky visibility over land from 1973 to 2007 is reported by Wang et al. (2009). This is interpreted in terms of aerosol concentrations and its impact on incident solar irradiance. A significant decrease is observed over Asia, South





America, Australia and Africa, while over Europe visibility increased after the 1980s, as a result of air pollution mitigation measures. Vautard et al. (2009) found a significant decrease in the frequency of low visibility days in Europe after the 1980s, which is spatially and temporally correlated with $SO_2$ emissions. Stjern et al. (2011) reported that emission reductions from 1983 to 2008 in the heavily industrialized area of central Europe (the formerly called Black Triangle, BT (named from the triangle of the meeting borders of Germany, Poland, and the Czech Republic) caused an increase of 15 km in the horizontal visibility, in contrast to the clean area where visibility increased by only 2.5 km. Doyle and Dorling (2002) observed significant improvement of visibility after early 1970s at many sites in UK, attributed to changes in the use of fuels, while Van Beelen and van Delden (2012) found that the proportion of days with high visibility (> 19 km) almost doubled since the early 1980s, in the Netherlands. These findings for Europe are in line with the so called dimming/brightening periods, referring to observed decreasing/increasing trends of surface solar radiation (SSR), associated with relevant changes in anthropogenic emissions (e.g. Streets et al., 2006; Wild, 2009; Cermak et al., 2010; Folini and Wild, 2011; Nabat et al., 2014).

In contrast to European areas, a tendency towards lower visibility is observed in developing countries (e.g. China, South Korea, South Taiwan, India), where it is still difficult to control air pollution (Ghim et al., 2005; Che et al., 2007; Wan et al., 2011; Singh and Dey, 2012; Wu et al., 2012). Along this line, Wu et al. (2012) found strong correlation between AOD and visibility in China over the period 2000-2009, and an overall decreasing trend in visibility (under sunny conditions) during the last 50 years. Singh and Dey (2012) correlated visibility in Delhi with aerosol composition and reported a rapid decrease of visibility during 1980-2000, and stabilization afterwards.

Urban environments are of particular interest, as air pollution from local sources is superimposed on other regional factors, strongly impacting visibility (Davis, 1991; Eidels-Dubovoi, 2002; Tsai et al., 2003, 2007; Dayan and Levy, 2005; Chang et al., 2009; Kim, 2015).

The present study explores the historical observations of visibility in Athens, which is the oldest time series of visibility in Greece and, to our knowledge, one of the oldest, uninterrupted time series of visibility in the eastern Mediterranean. The records are retrieved from the historical climatic archives of the National Observatory of Athens (NOA) and span a period of more than 80 years (1931-2013). In the past, Carapiperis and Karapiperis (1952) reported on the correlation between the visibility and the blue colour of the Attika sky, while Kanellopoulou (1979) analysed visibility in Athens for the period 1931-1977 and reported a pronounced decrease after the 1950s. Since then, there has been no other study addressing changes in visibility, as well as the factors





behind these changes, during the last 40 years, when significant changes occurred in Athens in terms of urban
expansion, traffic load, 2004 Olympic Games constructions and the economic recession (starting in 2008). The
inter-decadal variability and long-term trends of visibility in Athens are presented in the study. The role of
meteorology and aerosol loads (of local and regional sources) on the variability and trends are investigated and
discussed, while the relationship between visibility and aerosol loadings is investigated, through the analysis of
satellite AOD retrievals over Athens since 2000, but also surface measurements of PM10 in Athens and Finokalia
station (Crete) over shorter periods.

## 2 Study area and data

### 2.1 Study area

Athens, the capital of Greece, concentrates the largest part of the commercial, financial, societal and cultural
activities of the country. The Greater Athens Area (GAA) (Fig. 1) extends beyond the administrative municipal
city limits and covers a surface of 433 km$^2$. The population of GAA is approximately 3.7 million (almost twice
the population of 1961) and accounts for more than one third of the Greek population. The growth of the
population was coupled with the number of vehicles. Specifically, the number of private cars rose from 2 % of
inhabitants in 1964 to 44 % in 2008. The population growth and the increased number of automobiles has caused
traffic problems, increased anthropogenic emissions and degradation of air quality in the city. The complex
topography, consisting of relatively high mountains around GAA (Fig. 1), induces poor ventilation of the city.
Sea/land breezes appear along the axis NE - SW and have a major role in the accumulation of air pollutants
(Kalabokas et al., 1999a, b).
In order to compare our findings for Athens with a reference, remote site, the visibility records from the
Heraklion airport (HER) in Crete Island, were used (Fig. 1). Heraklion is located about 330 km south of Athens,
while its airport is 5 km east of the city with no significant (or systematic) influence from the urban web.

### 2.2 Climatic features of Athens

Athens has a temperate climate, with warm and dry summers and more wet and mild winters, typical for eastern
Mediterranean. Table 1 presents monthly and annual normal values along with standard deviations of the daily
mean, maximum and minimum air temperature, precipitation amount and precipitation frequency (PF) (defined


as the number of days with total precipitation > 1 mm, following WMO), relative humidity and wind speed in
Athens, based on the WMO reference period, 1971-2000. July and August are the warmest and driest months of
the year. Actually, the periods from May to September and from October to March represent the dry and wet
periods of the year respectively. Precipitation is sparse in summer (June- August), with the total amount
averaging 20 mm and precipitation frequency averaging 3 days. Athens receives on average approximately 400
mm of rain per year, corresponding to 43 rainy days (Table 1).
During summer, the area is dominated by anticyclonic circulation that enhances air temperature and intensifies
urban heat island. Athens has been experiencing a significant warming since the mid 1970's, more pronounced in
summer, which is the additive result of regional warming and gradual intensification of the urban heat island
(Founda, 2011; Founda et al., 2015). Strong northeasterly winds in late summer, known from antiquity as
'Etesians', induce a relief on air temperature and air pollution levels in the city.
Figure 2b presents the seasonal variability of air masses origin over Athens according to the sectors defined in
Fig. 2a, based on 10-yr climatology of daily air trajectories. The S (south) sector is linked to transport of air
masses from arid areas of N Africa, frequently associated with dust events that affect the eastern Mediterranea
(Hamonou et al., 1999; Gkikkas et al., 2015), the N (north) sector accounts for Balkans and the main continental
Europe, while the W (west) sector corresponds to SW Europe and the W Mediterranean Basin. Note that air
masses transport from the W sector are significantly blocked by the high altitude mountain chain of Pindus (>
2500 m), that expands from North to South along western Greek mainland. Air masses origin was identified by
applying a 4-day back-trajectory analysis, calculated daily at 12:00 UT with the Hybrid Single-Particle
Lagrangian Integrated Trajectory (HYSPLIT) model (version 4.9) (Draxler et al., 2009).
On an annual basis, air masses from the N and NE sectors dominate, contributing by more than 60 % and
showing profound seasonal variability (maximum in summer). Similar conclusions were obtained based on
surface wind speed and direction measurements reported in Fig. 3. Winds from N-NE directions prevail in Athens
at a frequency of nearly 38 % (Fig. 3). This sector is also associated with the occurrence of high wind speeds, as
shown in the same figure. The second most frequent surface winds correspond to S-SW directions (27%). The
frequency of occurrence of this sector maximizes during the intermediate seasons (spring and autumn) and is
associated with the occurrence of dust events from northern Africa and, in cases of light winds, with sea breezes
from the Saronic Gulf (Fig. 1).
**2.3 Overview of air pollution in Athens**



A short introduction on the factors that diachronically control air pollution levels in Athens is presented here, to facilitate the interpretation of visibility variations in terms of pollutants concentrations.

Air pollution in Athens has been systematically measured since the early 1970s. Road transport, domestic combustion and industrial activity have been the main sources of air pollution in GAA, throughout the years. Downward trends of sulfur dioxide, black smoke, carbon monoxide and nitrogen oxides have been reported from the mid 1980s to the late 1990s, attributed to several anti pollution measures adopted by the state (e.g. replacement of the old technology gasoline-powered private cars and the reduction of the sulfur content in diesel oil) (Kalabokas et al., 1999a). Negative trend of $NO_2$, NOx and $O_3$ from the mid 1980s to 2009 is also reported in several urban stations (Mavroidis and Ilia, 2012).

Measurements of particulate matter (PM) had been only occasionally conducted in Athens before the EU Directive (1999/30/EC) was launched, revealing increased concentrations of $PM_{10}$ (Hoek et al., 1997). Chaloulakou et al. (2003) reported on $PM_{10}$ and $PM_{2.5}$ at a single road traffic sampling location from 1999-2000 and underlined the contribution of local emission sources, mostly traffic, on the high levels of PM concentration. Grivas et al. (2004) highlighted the significant vehicular contributions in $PM_{10}$ concentrations in Athens during 2001-2004 and quantified the exceedances of the annual limit set by the EU Directive.

Studying the contribution of local sources versus regional and the role of long-range transport over megacities of the eastern Mediterranean, including GAA, Kanakidou et al. (2011) summarized that a significant number of PM exceedances registered in Athens, are associated with regional pollution sources or natural dust transport, clearly highlighting the importance of regional transport processes. Theodosi et al. (2011), compared simultaneous mass and chemical composition measurements of size segregated particulate matter ($PM_1$, $PM_{2.5}$ and $PM_{10}$) at two urban and a reference, non-urban background site, concluding that, during the warm season there is no significant (actually < 15 %) difference in $PM_1$ between the urban and reference sites, while on the other hand, local anthropogenic sources dominate during the cold season. Regarding the coarse fraction, a significant contribution from soil was found in urban locations throughout the year, contributing significantly (up to 33 %) to the local $PM_{10}$ mass.

Regarding columnar aerosol loads and using ground-based AOD measurements in Athens, Gerasopoulos et al. (2011) showed that the greatest contribution (40 %) to the annually averaged AOD, comes from regional sources (namely the Istanbul metropolitan area, the extended areas of biomass burning around the north coast of the Black Sea, power plants spread throughout the Balkans and the industrial area in the Po valley). Additional



important contributors are dust from Africa (23 %), whereas the rest of Europe contributes another 22 %. Gkikkas
et al. (2015) found good correlation between $AOD_{550nm}$ and surface $PM_{10}$ over the Mediterranean basin during
desert dust episodes (2000-2013) and reported higher intensity but lower frequency of such episodes over the
central and eastern Mediterranean. Additionally, Hatzianastassiou et al. (2009) found that local anthropogenic
emissions in GAA contribute by 15-30 % to the total AOD, as derived from satellite-based AOD measurements.
Vrekoussis et al. (2013) reported on the improvement of air quality in Athens during the period 2008-2013, as a
result of the economic recession and the subsequent cut down on vehicles use and industrial activity. For the
same period, Paraskevopoulou et al. (2014) showed that the massive turn of Athens' population to wood burning
for residential heating purposes gave rise to smog episodes characterized by high PM spikes during night time in
winter. A longer-term (2008-2013) analysis of aerosol chemical composition and sources at a suburban site in
Athens by Paraskevopoulou et al. (2015) revealed that the area of Athens is now generally dominated by aged,
transported aerosols.
**2.4 Visibility observations in Athens**
The historical climatic records of the National Observatory of Athens (NOA) were used in this study. NOA is
established on the Hill of Nymphs (latitude: 37.97 $^0$N, longitude: 23.71 $^0$E, altitude: 107 m, above sea level), at
the historical center of the city, near Acropolis. The location of the observations on the top of a hill ensures
unobstructed view towards all directions. Visibility observations have been conducted uninterruptedly at NOA at
least 3 times per day, since the late 1920's. Daily observations of visibility at 14:00 LST (LST = UT + 2hrs),
from 1931 to 2013 were used in the study. The time series is complete, with a very short gap of 6 days occurring
in December 1944, owed to political convulsion in the country at that period.
Visibility data at other stations (e.g. Heraklion, Crete) were extracted from the network of the Hellenic National
Meteorological Service (HNMS) and actually represent visibility observations at the airport station, initiated after
mid the 1950s. Meteorological data for Athens over the period 1931-2013, were also acquired from the historical
archives of NOA. Monthly, seasonal and annual mean values of visibility were derived from the daily
observations at 14:00 LST.
An empirical scale of visibility classes, as recommended by the World Meteorological Organization (WMO), has
been used for visibility observations at NOA (Table 2). Classes are defined based on the greatest distance at
which a predefined object can be seen and recognized with naked eye. The procedure requires that an operator





scans the horizon for predetermined objects. In the case of Athens, some historical buildings in the city, but also
certain objects of the surrounding landscape, unaltered over the years, (e.g. objects on the mountains or islands of
the Saronic Gulf, Fig. 1), were chosen to represent visibility classes and relevant distance ranges. The procedure
inevitably introduces some kind of subjectivity and bias in the measurements, related to individual eyesight of
different operators. It is assumed however, that the execution of visibility observations by different operators over
the years could have possibly had a compensating effect and an overall reduction of biases. More details about
the possible errors and validity of visibility observations have been thoroughly discussed by Davis, (1991).
The use of the WMO scale introduces a further uncertainty on visibility observations, associated with the
amplitude of visibility ranges corresponding to each visibility class. Information on the use of WMO scale and
relative uncertainties, as well as the procedure followed for averaging daily visibility observations is provided in
Supplementary materials.

## 2.5 Aerosol data used in the study

Long time series of atmospheric pollution measurements in Athens and the selected reference site would enable
drawing direct relationships between visibility and aerosols and would provide evidence on the character
(regional or local) of atmospheric pollution in Athens and its impact on long-term visibility variations. Given that
such time series are missing, we used shorter time series of aerosol measurements for a direct comparison
between visibility and atmospheric pollution in Athens.
In an effort to explore the relationship of visibility with AOD over Athens, we used the Terra/Modis AOD at 550
nm, available since 2000. NASA's Terra satellite is sun synchronous and near polar-orbiting, with a circular orbit
of 705 km above sea level. MODIS is capable of scanning 36 spectral bands across a swath 2330 km wide.
MODIS aerosol products were used in order to analyze the temporal and spatial variability of aerosols over the
wide area of interest. In this study, we used daily level-2 collection 5.1 MODIS/Terra AOD at 550 nm. Daily
overpass data for the specific area were extracted at a spatial resolution of 50 x 50 km$^2$. Previous studies have
shown that such special resolution product ensures sufficient daily measurements without losing out to the higher
spatial resolution and hence provide a better opportunity of correctly viewing the atmospheric aerosol load
(Ichoku et al., 2002). The overpass time is $09:35 \pm 45$ min UT.
Surface PM10 measurements in Athens were also used to verify the relationship between visibility and
particulate pollution from surface measurements. It is well known that desert dust plumes are often transported in


altitude over the Mediterranean (e.g. Hamonou et al., 1999: Gkikkas et al., 2015) and a portion of surface PM
exceedances in Athens is associated with natural dust transport (Kanakidou et al. (2011). The analysis was based
on a short data set of $PM_{10}$ measurements at two stations in Athens (Aristotelous and Maroussi), covering the
period 2008-2012. Aristotelous is an urban street station in the center of the city and Maroussi is a suburban
station, at a distance of about 15 km to the North of NOA.
Finally, a data set of $PM_{10}$ measurements at a reference station in Crete (Finokalia station), covering the period
2005-2014 was used, for the detection of any trends, representative of regional atmospheric pollution trends.
Finokalia station is located at a distance of less than 50 km East of Heraklion airport.

**3 Results**
**3.1 Inter-decadal variation of visibility and trends**
Figure 4 displays the long-term evolution and variability of the annual visibility in Athens from 1931 to 2013.
The population growth in the city of Athens over the same period is also shown, while the figure also displays the
long-term variability of the relative humidity in Athens (which is discussed below). It is obvious that the annual
visibility in Athens has undergone a very strong and almost continuous decline over the past 80 years, in
coincidence with the increase in population. The long-term linear trend over the whole studied period was found
to be equal to -2.8 km decade$^{-1}$ (p < 0.001). However, this trend is not constant throughout the entire studied
period. Three sub-periods are visually discerned in Fig. 4 (also confirmed with sensitivity tests): (a) 1931-1948,
(b) 1949-2003 and (c) 2004-2013. Visibility levels are remarkably higher in the first sub-period varying around
25 km. A slight negative trend is observed during this period    (-0.66 km decade$^{-1}$). In the late 1940s, visibility
experienced a striking and abrupt decrease at the time of population first burst, which was then followed by a
progressive deterioration, at least until the early 2000s. In this second sub-period (1949-2003) visibility decreases
at a rate of -2.33 km decade$^{-1}$ (p < 0.001). A tendency of stabilization or even recovery seems to be established
during the recent decade 2004-2013, with visibility showing a slight increasing trend (+ 0.07 km yr$^{-1}$). A detailed
discussion on the observed trends and their linkage with air pollution is presented in section 3.5.

**3.2 Frequency distribution of visibility ranges**



Figure 5 illustrates the frequency of occurrence of different visibility ranges as described in Table 2 for the three
sub-periods. In the first sub-period, visibility values lie within the range of 10-20 km at a percentage of 36 % and
of 20-50 km at a percentage of 34 %. Very high visibility (>50 km) accounts for a considerable percentage (~9 %)
and poor visibility (< 2 km) corresponds cumulatively to only 2 %. The frequency of visibilities lower to 1 km is
very low (0.4 %), while visibility was found to be lower to 500 m only in 9 cases. Cumulatively, visibility
exceeding 10 km corresponds to approximately 80 % of the cases during this period.
A shift towards lower visibility values is observed during the second sub-period, namely 1949-2003. Specifically,
the most frequent visibility ranges are 4-10 km (38 %) and 10-20 km (34 %). The frequency of visibility > 50 km
is negligible (0.6 %) and the frequency of poor visibility (< 2 km) amounts cumulatively to 5.6 % , with 0.9 %
corresponding to visibility < 1 km. Visibility lower to 500 m was observed only in 12 cases. Cumulatively, the
percentage of days with visibility exceeding 10 km drops to 45 % during this sub-period.
The frequency distribution changes dramatically during the most recent period (2004-2013). In particular,
although visibility range of 4-10 km remains the most frequent (30%) as in the second sub-period, almost similar
frequency (~28 %) is also observed in the range of 2-4 km, corresponding to a doubling of the percentage of this
category. The frequency of poor visibility (<2 km) rises to approximately 25 %, with a substantial percentage (5.6
%) accounting for visibility lower to 1 km and 0.46 % lower to 500 m. Cumulativelly, visibility did not exceed 4
km for half of the days of the year during 2004-2013. The percentage of days with visibility > 10 km is 18%,
while frequency of very good visibility ( > 20 km) amounts to just 2 %. No case of visibility > 50 km was
observed in this sub-period.
**3.3 Seasonal variation of visibility**
Since visibility is influenced by the prevailing meteorological conditions (Davis 1991; Sloane 1982), it is
expected that it will also exhibit a seasonal variability, depending on the intra annual variability of climatic
conditions at the examined area. Mean monthly values of visibility were calculated for all three sub-periods.
Figure 6 presents the mean monthly values of visibility in Athens over the three sub-periods, normalized with the
value of the month with the highest visibility. In the same plot, the mean monthly values of the relative humidity
(RH), coinciding visibility observations at 14:00 LST over the period 1931-2013, are also shown. It is noteworthy
that RH at NOA does not exhibit any significant trend over the years (as already shown in Fig. 4) and its monthly
distribution is almost unaltered over the years. As it comes out from Fig. 6, visibility shows a distinct seasonal
cycle in all three sub-periods, with better visibility occurring in the warm and dry season of the year. Although





seasonality is observed in all sub-periods, the pattern is more evident and robust in the first sub-period, with
much higher visibility values (up to 40%) in the warm and dry months compared to cold and wet months. The
pattern of visibility in this period is almost a mirror image of the pattern of RH and reflects the influence of RH
on visibility and the anti-correlation between these two variables. The lowest values of RH correspond to July
and August (mean value of RH ~35%) and this probably results to improvement of visibility. Moreover, strong
northeastern winds (the so called 'Etesians') that prevail in eastern Greece during these months enhance
ventilation and induce drier conditions in the city, therefore improving visibility.
In the other two sub-periods, 1949-2003 and 2004-2013, higher visibility values are also observed during the
warm and drier months (Fig. 6), however, the distinct seasonal cycle observed in the first sub-period has changed.
During the second sub-period in particular, seasonality is noticeably attenuated and visibility differences between
the warm and cold period is of the order of 10%. This possibly implies a weakening of the influence of
meteorological conditions as a result of (or in combination with) stronger effect of air pollution on the visual air
quality of the city.
The minimum of visibility is constantly observed in March during all sub-periods. Indeed, March falls in the
transitional season of the year and thus bears higher values of RH compared to summer months (mean value of
RH at 14.00 LST > 50 % and mean daily value 67 % in March). Additionally, March falls in the growing season,
with enhanced pollen and biogenic aerosol emissions which is a known factor for visibility impairment (e.g. Kim,
2007). Increased frequency of dust outbreaks from northern Africa in spring, influence extensively the area of
eastern Mediterranean (Hamonou et al., 1999; Gerasopoulos et al., 2005, 2011; Gkikkas et al., 2015) and thus
constitute a major factor for visibility impairment during spring months. Léon et al (1999) reported that ~ 40 % of
the days with high aerosol optical depth at 865 nm ($AOD_{865nm} >$ 0.18) over Thessaloniki (Greece) were
associated with African dust transport events, all observed in the period March – July, while Dayan and Levy
(2005) found higher $PM_{10}$ values and lower visibility levels during spring in Tel Aviv, associated with the
frequent passage of cyclones that cause natural dust outbreaks.
**3.4 Visibility and meteorological conditions**
The impact of meteorological conditions on visibilty has been investigated by different researchers using
different approaches , as for instance the classification of synoptic circulation patterns (Sloane, 1982; Davis,
1991; Dayan and Levy, 2005), the application of correction factors on extinction coefficient to account for RH
effect (Che et al., 2007), the estimation of correlation coefficients between visibility and meteorological variables





(Deng et al., 2011), or simply the comparison of diurnal /seasonal cycles and temporal trends of visibility with
the relevant cycles and trends of meteorological variables (Van Beelen and van Delden, 2012). Sloane (1982)
reported that periods with exceptionally maxima or minima of visual air quality were related (apart from sulphate
emissions) with favourable synoptic circulation patterns. Studying visibility in Tel Aviv (Israel), Dayan and Levy
(2005) reported a strong dependence of visibility levels from meteorological conditions, synoptic weather
patterns and air mass origin, with the highest mean values occurring in summer, related to the persistent nature of
the summer synoptic weather pattern in the eastern Mediterranean. Deng et al. (2011) found that RH and wind
speed were significantly correlated with visibility at an urban area of China, while Ghim et al. (2006) showed a
considerable decrease in visibility in South Korea, despite the observed simultaneous decrease of the relative
humidity levels. The relationship and possible impact of different meteorological parameters such as
precipitation, RH, wind speed and wind direction on visibility in Athens is discussed below.

### 3.4.1 Visibility and precipitation

Precipitation is associated with scavenging of atmospheric particles (e.g. Remoudaki et al., 1991a; 1991b),
possibly resulting to improvement of visibility. The precipitation frequency in particular, was found to control
seasonal variability of the total atmospheric deposition of lead in western Mediterranean (Remoudaki et al.,
1991b). Rainy days on the other hand are associated with increased relative humidity, resulting in reduction of
visibility. A plot illustrating the long-term variability of the annual precipitation amount and precipitation
frequency (PF) at NOA from 1931-2013 was created, for the detection of any significant temporal trends which
might have an effect on visibility trends (Fig. 7). According to Fig. 7, no long-term trend is observed in the
annual precipitation amount at NOA from 1931-2013, which could have had an effect on long-term trends of
visibility. Precipitation frequency on the other hand exhibits an overall negative trend over the same period (-1.1
days decade$^{-1}$), not constant, though.  Actually, PF decreases from the late 1960s to the late 1980s, while it
presents an increasing tendency after 1990 (+1.3 days decade$^{-1}$).  The correlation coefficient between annual
visibility and PF was found to be positive only during the period from early 1970s to the late 1980s (+ 0.45, $p <$
0.05). A negative correlation coefficients was found in the post 1990 period (-0.21), not statistically significant.
Subsets of data were also produced for the creation of additional visibility time series, accounting for
precipitation influence. Figure 8 presents visibility variability during the wet (October-March) and dry (May-
September) period of the year, along with the annual values. Lower values during the rainy and cold period of the
year are most probably associated with higher values of relative humidity, resulting to reduction of visibility.





Despite the differences between the time series in Fig. 8, the overall tendency is similar, thus not affecting the
validity of our conclusions as regards long-term visibility impairment in Athens. Additional plots created from
subsets of 'rain' and 'no rain' days are provided in Supplementary materials (Fig. S4).

### 3.4.2 Correlation between visibility and other meteorological parameters (RH, wind)

Figure 9 presents the running correlation coefficient (15-yrs window) between visibility and relative humidity,
over the period 1931-2013. As expected, the correlation coefficient between visibility and RH is negative,
indicating the anti-correlation between these two variables. High RH enhances water uptake by airborne particles,
leading to higher light scattering and thus visibility impairment. Actually, when RH exceeds a threshold level
(e.g. > 70%) some inorganic salts, such as ammonium sulfate and nitrate, undergo sudden phase transitions from
solid particles to solution droplets and become disproportionately responsible for visibility impairment, as
compared with other particles that do not uptake water molecules (Malm, 1999).
As it comes out from Fig. 9, the negative correlation between RH and visibility is statistically significant ($p <$
$0.01$) almost over the entire studied period. However, a progressive weakening of the correlation coefficient with
time is observed, indicating a less strong correlation between the two variables over the years. Stronger anti-
correlation is found until early 1970s, followed by lower (still significant) values till late 1970s. The progressive
weakening of the correlation between RH and visibility in Athens, possibly suggests a progressive weakening or
mask of the influence of RH on visibility, compared to the effect of other factors such as atmospheric pollution
(although the influence of RH is enhanced in the presence of certain hygroscopic particles). On the contrary, the
impact of surface wind speed on visibility seems to be stronger during the recent decades (Fig. 9). Higher wind
speeds in this case (positive correlation) are related to the dispersion of air pollutants and the more efficient city
ventilation. In others cases wind speed is also used as a proxy for long-range transport, but then a negative
correlation would be expected. Lower values of the coefficient in the first decades possibly demonstrate that the
lack of pollutants at that period diminishes the importance of ventilation. The correlation coefficient
progressively increases over the years. The rate of increase is higher after the mid 1980s, when correlation
becomes statistically significant ($p < 0.01$). Similar values (~ 0.29) of correlation coefficient between light
extinction coefficient and wind speed are reported by Deng et al. (2011) in China.
Apart from wind speed, visibility was also found to be sensitive to wind direction. A distinct variability of
visibility with wind direction is observed in Fig. 10, for all sub-periods. Lower values of visibility are related to
southerly winds, as they either bring dust from Sahara or warmer and more humid air masses from the sea (see



also Figs 1, 2b). Southeasterly winds are in general weak winds (see Fig. 3), while southwesterly winds are
associated with sea breezes from the Saronic Gulf (Fig. 1). In general, sea breeze and calms favor the
accumulation of pollutants, the formation of secondary aerosols and photochemical smog in Athens (Colbeck et
al., 2002), thus reducing visibility. A number of S/SW events are also associated with strong wind speeds
occurring during Sahara dust outbreaks, which enrich Athens atmosphere with dust particles that decrease
visibility (Figs 2, 3). As it comes out from Fig. 10, the highest visibility occurs under northwesterly winds and
this is robust over the entire studied period. An explanation for this, is that air masses originated from
northwesterly directions are much drier as they have lost water vapor after passing over the high mountainous
basin of Greek mainland (e.g. Pindos mountain), while air pollution is also blocked within the boundary layer by
the mountain chain.
**3.5 Air pollution and urbanization relations to visibility**
In this section we attempt to interpret the observed inter-decadal variability and trends of visibility in Athens, in
terms of air pollution. As already shown in Fig. 4, the pre-1950 period is characterized by much better visibility
in Athens. From then on, visibility experienced a rapid decrease, followed by a smoother but continuous negative
trend until the early 2000s. The period after 1950 signifies the post World War II epoch but also coincides with
the end of a civil war in Greece (1946-1949), which was followed by an important urbanization wave in Athens
(Maloutas, 2003). This is in line with the growth of Athens' population, as illustrated in Fig. 4. The greatest rate
of population increase is observed between 1950 and 1960, when population in Athens almost doubled. The
population growth was associated with a significant increase of constructions in the city. Apart from the intense
urbanization in Athens, this period is also characterized by the most prominent increase of anthropogenic
emissions on a global and European scale (e.g. Mylona, 1996; van Aardenee et al., 2001), which is discussed
below.
Although in the second sub-period, 1949-2003, visibility was found to be remarkably lower compared to the first
one, a slight recovery of visibility was observed during the recent decade, 2004-2013 (Fig. 4). This improvement
could be related to a number of reasons. The years after 2004 correspond to the post Olympic Games period in
Athens. A number of important transport projects were completed prior to the Olympic Games in Athens in 2004.
Such projects are for instance the construction of the Attika Ring Road (one of the largest in Europe), the
construction of Tramway and the extension of Athens Metro. These projects have contributed to the reduction of
the number of vehicles in the city, resulting to less traffic problems and lower air pollution levels. Another



possible contributing factor concerns the possible impact of the Greek economic recession (2008-2013) on air
quality in Greece, and Athens in particular. Recent studies provide some evidence for this. For instance,
Vrekoussis et al. (2013) found strong correlation between different economic metrics and air pollutants after
2007, suggesting that the economic recession has resulted in proportionally reduced levels of air pollutants in the
two biggest cities in Greece. This is further supported by other recent research studies that report a significant
reduction in energy consumption after 2008, related to the rapid economic degradation (Santamouris et al., 2013).
But how far are these changes in visibility in Athens due to local factors or can be considered representative of a
more extensive area? To answer this question and also evaluate our findings as regards the urban influence, the
Athens visibility record is compared with visibility at a reference, non urban station. From the available stations
in Greece disposing long-term visibility observations, we chose the station at Heraklion airport (HER) in Crete
Island. Actually, both sites, NOA and HER, are most of the year exposed to air masses of similar origin (from
northeasterly directions), travelling over the Aegean Sea, in contrast to other sites of the country that are strongly
affected by the mountainous volumes of the Greek mainland. Visibility observations at HER are available since
the mid 1950s. Figure 11 presents the long-term variation of the annual visibility at HER along with annual
visibility at NOA. Linear trends of the two time series for their common period (1956-2009) are also shown in the
figure. The time series were found significantly correlated (correlation coefficient>0.88, p<0.05).
As it comes out from Fig. 11, visibility levels at urban NOA are constantly lower by a few km (~ 7 km) compared
to the background station, HER. It is remarkable that during the first two decades of parallel observations, both
curves show significant covariance, easily realized from the peaks in 1959, 1966 and 1970 and the minima in
1963 and 1973, suggesting the impact of large scale phenomena (for instance, volcanic eruptions in 1963) in the
modulation of visibility levels. A prominent feature in Fig. 11 is that the background visibility at the reference
site has been also on a downward route since the mid 1950s, in accordance to the observed decreasing trend of
the visibility in Athens. As already stated, the beginning of the 1950s signifies a period with an outstanding
increase of emissions in Europe. European $SO_2$ emissions in particular, increased almost at a constant rate during
the first half of the 20th century, while they experienced a quite abrupt increase in the 1950s and almost doubled
their values between 1950 and 1960 (van Aardenne et al., 2001; Mylona, 1996). Figure 11 includes the rates of
$SO_2$ increase per decade in Europe (in Tg S decade$^{-1}$), as reported by van Aardenne et al. (2001). Constant
increasing rates (2 Tg S decade$^{-1}$) are observed untill 1950, when the rate of increase reached 6 Tg S decade$^{-1}$
between 1950-1970. A decline of the increasing rate is then observed, while in the 1990s European sulfur
emissions stabilize. Stabilization of emissions is followed by a continuous decline after 1990. Stjern et al. (2011)





reported a prominent decrease of $SO_x$ emissions and sulphate in aerosols in both eastern and western Europe from
1990-2007, but with higher rates of decrease in eastern Europe.
A very important finding in Fig. 11 is the similar slopes in the linear trends of the annual visibility at the
background and urban stations, over their common period of observations (-2.2 km decade$^{-1}$ and -2.4 km decade$^{-1}$
, respectively). This feature implies that, apart from the absolutely lower values of visibility in the urban web of
Athens, the inter-decadal variability of visibility in the city and the extended area is significantly modulated by
large scale processes that control regional visibility, such as long-range pollution transport and/or changes of
atmospheric circulation. Many studies have identified the eastern Mediterranean as a crossroad of aerosols of
different origins, sizes and chemical composition (Lelieveld et al., 2002; Hatzianastassiou et al., 2009; Kanakidou
et al., 2011; Gerasopoulos et al., 2011), which inevitably affect optical properties of the atmosphere. Kanakidou
et al. (2011) found that even in the large urban regions of the eastern Mediterranean, particulate matter has a
significant contribution by distant anthropogenic pollution sources in the region but also by long-range transport
of African dust.
After the early 1990s, the time series diverge, with background visibility partly recovering, and visibility in
Athens keeping declining at the same pace until 2003 (Fig. 11). Recovering of visibility at other Greek areas
around the 1990s is also found by Lianou et al. (unpublished data) which is also in line with the observed
visibility improvement in other European areas, related to emissions reduction (Wang et al., 2009; Vautard et al.,
2009). This last feature suggests that during this period, local emissions might have a dominant role in the
determination of visibility in Athens.
**3.6 Visibility in Athens and AOD**
The realtionship of visibility with AOD over Athens was also explored using satellite data since 2000 (see
Section 2.5). The AOD time series showed a significant (-2.4% per year) decrease from 2000 up to 2010 and a
further decrease of (-7.4% per year) for the 2010-2014 period (Fig.12).
To investigate the relationship between visibility and AOD changes, the two parameters are plotted together after
data binning. Visibility and AOD measurements have been used as follows: Visibility at 12:00 UT was used
according to the indices defined in Table 2 and plotted against average AOD from synchronous satellite
overpasses. The mean AOD and its standard deviation are presented in Fig. 13. The AOD values are related with
the visibility data using as the distance in km the middle point of each visibility bin (range). Only summertime



(June-August) MODIS AOD have been used, to keep visibility values unaffected from other atmospheric
parameters like low clouds, rain or relative humidity. It is observed that for average AOD values for Athens (0.25
using the mean June-August AOD at 550nm from our MODIS AOD dataset or 0.23 at 500 nm as reported by
Gerasopoulos et al., 2011) visibility varies in the range of 4 km to 10 km. For cleaner conditions (W-NW-N, 0.12
- 0.17 at 500 nm, Gerasopoulos et al., 2011) visibility can go as high as 20 km, while very low visibility (< 0.5
km) is generally associated to the highest aerosol loads, with AOD > 0.3 (e.g. in the case of dust events, long-
range transport of urban/industrial pollutants and stagnant conditions).
**3.7 Visibility in relation to $PM_{10}$**
An additional analysis was conducted to verify the relationship between visibility and particulate pollution from
surface measurements using a short data set of $PM_{10}$ in Athens as described in Section 2.5. Figure 14 presents
visibility variation as a function of $PM_{10}$ levels measured at Aristotelous (urban) and Maroussi (suburban)
stations. Four different classes of $PM_{10}$ levels were used, as shown in Fig. 14. The frequency of occurrence of
each class is also shown in the figure. Despite the different locations and characteristics of the two stations, the
observed frequencies are very similar in all classes of $PM_{10}$ levels, with higher frequency corresponding to the
class of 30 -60 μg m$^{-3}$ at both stations. The frequency of $PM_{10} > 90$ μg m$^{-3}$ at Aristotelous is double compared to
the respective frequency at Maroussi. Independently of the location, the same strong relationship is observed
between visibility reported at NOA and $PM_{10}$ levels at both stations, revealing a prominent decrease of visibility
with increasing $PM_{10}$ levels, in agreement with our conclusions. Average visibility at NOA ranged between 8 and
9 km under low $PM_{10}$ levels (< 30 μg m$^{-3}$), but is reduced to less than 3 km under severe episodes of particulate
pollution ($PM_{10} > 90$ μg m$^{-3}$). The correlation coefficient between daily measurements of $PM_{10}$ levels and daily
visibility at NOA was found equal to -0.38 (p < 0.05) and -0.36 (p < 0.05) for Aristotelous and Maroussi sites
respectively.
Figure 15 displays the variation of the mean annual values of $PM_{10}$ at the reference station of Finokalia (Crete)
over the 10-yr period (2005-2014), along with standard deviations. A decreasing tendency in $PM_{10}$ levels is
observed, which is also consistent with the slight recovery of visibility levels in Athens over the same period.

**4 Discussion and Conclusions**




The present work analyses for first time the historical record of visibility in Athens (NOA) from 1931 to 2013
and explores its long-term variability and trends. An attempt was made to interpret the temporal variations of
visibility in terms of relevant changes of atmospheric properties (related to local or regional processes) and/or
meteorological conditions. Since this is the longest record of visibility observations in Greece and one of the
oldest in the broader area of the eastern Mediterranean, the analysis provided valuable information on the
atmospheric properties of the area in the past, when air pollution records were missing.
The study period was divided into sub-periods corresponding to different visibility trends in the time series, each
sub-period being affected by different factors.
The role of meteorology on visibility was investigated in different ways. Visibility in Athens was found to reveal
a distinct seasonal cycle, with higher visibility corresponding to the warm and dry months of the year (namely
from May to September) and lower to the colder and wet months. Seasonality is more evident in the first sub-
period, when visibility in summer is up to 40% larger compared to winter. After the 1950s, the seasonal cycle
attenuates and the differences in visibility between summer and winter months were found to be much less
pronounced (of the order of 10%, Fig. 6). Lower visibility values were observed in March in all sub-periods,
resulting from the combination of enhanced pollen and biogenic aerosols emissions, but also to increased dust
outbreaks from northern Africa and relatively higher RH levels.
As expected, visibility was found to be negatively correlated with RH, but correlation is stronger in the first sub-
period and attenuates over the years. On the contrary, a positive correlation between visibility and wind speed
was detected which is statistically significant ($p < 0.01$) only during recent decades. Actually, stronger winds
seem to improve visibility as they induce a cleanup of the atmosphere from air pollutants.
Visibility was found to be very sensitive to wind direction, reflecting the influence of air masses origin.  Lower
visibility levels are constantly observed under southerly winds (Fig. 10). Such winds correspond to sea breeze
circulation associated with increased humidity levels but also to accumulation of air pollutants in the city and
formation of secondary air pollutants. In addition, some S/SW events are associated with strong wind speeds
(Fig. 3) occurring during Sahara dust outbreaks. These events enrich Athens with airborne particles, thus
decreasing visibility.
The study demonstrated that visibility in Athens has undergone a prominent impairment since the early 1930s.
The overall trend of annual visibility averages amounts to -2.8 km decade$^{-1}$.  The impressively higher levels of
visibility in Athens before the 1950s (also characterized by strong seasonality) reflect the transparency of the





atmosphere at that period, inherent to poorer aerosol loads from anthropogenic emissions (urban and/or regional).
The dramatic decrease of the visual air quality in the 1950s coincides with a number of events (end of wars, rapid
urbanization, increased emissions on local and regional scale) and points to the prominent role of aerosol loads in
the atmosphere of Athens. Air pollution has gradually incurred a severe visual pollution in the city, with visibility
lower to 4 km observed during more than half of the year in the recent decade, 2004-2013. The significant
decrease of visibility in Athens was not accompanied with analogous significant trends in RH or precipitation
(Figs 4, 7).
The comparison of the annual visibility in Athens with visibility at a reference, non urban site (HER) in Crete,
revealed some very interesting features. First, visibility in Athens was found to be constantly lower compared to
HER, possibly suggesting the impact of local anthropogenic emissions in the urban web. However, both time
series revealed similar and significant negative trends over their common period of observations (after the mid
1950s), pointing to the major contribution of long and regional range transport of natural and anthropogenic
pollution sources in the GAA urban area. Visibility deterioration after the mid 1950s is also reported in most
European areas, followed by stabilization and/or improvement around the 1980s or later (Vautard et al., 2009;
van Beelen and van Delden, 2012; Stjern et al., 2011). An improvement of visibility at HER around the 1990s
was not associated with analogous improvement of visibility in Athens, where visibility deterioration continued
until the early 2000s (Fig. 11). At that period, negative trends of main gaseous air pollutants are reported in
Athens (Kalabokas et al., 1999a). However, the direct effect of such pollutants on light extinction is negligible
compared to suspended particles and particularly to fine particles ($< 1\mu m$).
As already stated in Section 2.3, the contribution of both local and distant emission sources in PM concentrations
in Athens is suggested by a number of studies (e.g. Kanakidou et al., 2011; Gerasopoulos et al., 2011). Mainly
local emission sources (e.g. traffic) have been found to contribute to $PM_{10}$ concentration (Chaloulakou et al.,
2003; Grivas et al., 2004), while local anthropogenic sources seem to control $PM_1$ concentration only during the
cold months of the year (Theodosi et al., 2011). Using satellite-based AOD measurements, Hatzianastassiou et al.
(2009) found that local anthropogenic emissions in GAA contribute up to 30% to the total AOD.
A strong anticorrelation was found between visibility at NOA and $PM_{10}$ levels, measured at two different stations
(urban and suburban) in Athens over the period 2008-2012 (Fig. 14). The relationship between AOD and
visibility in Athens was examined in the study (Figs 12, 13). Illustrating the relationship between AOD, which
consist in a vertically integrated parameter, and visibility, a horizontally integrated parameter, requires various
assumptions. Using satellite based AOD and visibility observations for GAA, when assuming a vertically





constant extinction coefficient and a mixing layer that contains all aerosol load we end up describing the
theoretical relationship (Koschmieder, 1924): $Vis = k / AOD$, where $k$ is a function of the mixing layer height.
The 82-years long time series of visibility in Athens unfolded for first time information on the atmospheric
conditions in the area, for periods when atmospheric pollution measurements are missing. Although the analysis
is subject to several limitations and assumptions, mainly related to methods of visibility observations, the results
are robust and statistically significant, as the outstanding degradation of the visual air quality in the city over the
years.
The observed stabilization (or even slight improvement) of visibility in Athens in the very recent years could
possibly be related to reduced local anthropogenic emissions as a result of important transport infrastructures
(executed in view of Olympic Games) but also of the economic crisis in Greece. Although this last argument is
already supported by some recent research studies (e.g. Vrekoussis et al., 2013; Santamouris et al., 2013), the
impact of the economic crisis on local emissions seems to be more complicated and drawing out conclusions
remains tentative. Besides, in the same period regional atmospheric pollution presents a decreasing tendency, as
reflected in the negative trend of $PM_{10}$ levels measured at the background station of Finokalia in Crete (Fig. 15)
which is also consistent with the recent recovery of visibility in Athens.

**Acknowledgments.** The study is a contribution to the ChArMEX (The Chemistry-Aerosol Mediterranean
Experiment) work package on variability and trends. The study was supported by the Excellence Research
Program GSRT- Siemens (2015-2017) ARISTOTELIS "Environment, Space and Geodynamics/Seismology
2015-2017" in the framework of the Hellenic Republic-Siemens settlement Agreement. The authors are grateful
to the Editor Dr. François Dulac, for his very useful comments and suggestions on this study. The authors would
like also to thank the Hellenic National Meteorological Service (HNMS) for the provision of visibility data at
Heraklion (Crete) and the Air Quality Department of the Ministry of Environment & Energy of Greece for the
provision of air pollution data. The contribution of Mr. F. Pierros (NOA) and Mrs D. Koutentaki (NOA) in the
digitisation of visibility data of NOA and of Dr. G. Kouvarakis (University of Crete) in the analysis of air
trajectories is also acknowledged.

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




Table 1: Mean monthly and yearly values with standard deviations of basic climatic elements in Athens (NOA),
calculated from the WMO climatic period (1971-2000). (**)

| Month | Tmean ($^0$C) | Tmax ($^0$C) | Tmin ($^0$C) | RH (%) | Rainfall (mm) | Number of rainy days (> 1 mm) | Wind Speed (m s$^{-1}$) |
|---|---|---|---|---|---|---|---|
| January | 9.3 ±1.1 | 13.0 ±1.3 | 6.6 ±1.1 | 72.1 ±3.9 | 42.5 ±31 | 5.6 ± 3.0 | 3.1 ± 0.71 |
| February | 9.6 ±1.4 | 13. 7 ±1.7 | 6.8 ±1.4 | 70.2 ±3.5 | 44.8 ±29 | 5.6 ± 2.1 | 3.4 ± 0.50 |
| March | 11.5 ± 1.4 | 16.1 ± 1.8 | 8.2 ± 1.3 | 67.6 ± 4.3 | 50.2 ± 41 | 5.4 ± 2.6 | 3.3 ± 0.72 |
| April | 15.4 ± 1.3 | 20.5 ± 1.6 | 11.5 ± 1.1 | 62.7 ± 4.6 | 32.7 ± 29 | 4.2 ± 2.6 | 2.8 ± 0.51 |
| May | 20.3 ± 1.1 | 25.7 ± 1.3 | 16.1 ± 1.1 | 57.3 ± 4.0 | 16.7± 16 | 2.6 ± 1.9 | 2.9 ± 0.45 |
| June | 25.0 ± 0.9 | 30.6 ± 1.2 | 20.4 ± 0.9 | 51.3 ± 3.7 | 7.5 ± 10 | 0.9 ± 1.0 | 3.1 ± 0.60 |
| July | 27.3 ± 1.1 | 33.1 ± 1.4 | 22.7± 1.1 | 48.5 ± 4.2 | 6.6 ± 9 | 0.9 ± 1.1 | 3.5 ± 0.75 |
| August | 26.8 ± 1.2 | 33.7 ± 1.4 | 22.5± 1.2 | 49.8 ± 5.1 | 7.2 ± 12 | 0.9 ± 1.2 | 3.5 ± 0.58 |
| September | 23.4 ± 1.1 | 29.2 ± 1.5 | 19.4 ± 1.0 | 57.0 ± 4.7 | 9.4 ± 1 | 1.3 ± 1.6 | 2.9 ± 0.47 |
| October | 18.5 ± 1.5 | 23.5 ± 1.8 | 15.1± 1.6 | 66.4 ± 3.7 | 42.9 ± 40 | 3.7 ± 2.4 | 2.9 ± 0.74 |
| November | 14.0 ± 1.3 | 18.1 ± 1.5 | 11.1± 1.3 | 72.7 ± 3.8 | 59.9 ± 45 | 7.9 ± 3.8 | 2.9 ± 0.73 |
| December | 10.8 ± 1.4 | 14.4 ± 1.8 | 8.2 ± 1.3 | 74.0 ± 3.2 | 62.6 ± 34 | 9.0 ± 13.4 | 3.0 ± 0.56 |
| Year | 17.7 ± 0.5 | 22.6 ± 0.7 | 14.1 ± 0.5 | 62.0 ±1.9 | 389.5± 5 | 42.9 ± 9.0 | 3.1 ± 0.36 |

(**) Climatic means were calculated from daily observations at NOA over the period 1971-2000. Daily time series are
almost complete, with sporadic missing data in certain variables. In particular, data availability for the period 1971-200
equals 100 % for Tmax, Tmin and rainfall, 99.9 % for Tmean, 99.8 % for RH and 99.4% for the wind speed.





Table 2: The WMO empirical scale for visibility observations, used at NOA.

| Visibility Classes | 1 | 2 | 3 | 4 | 5 | 6 | 7 | 8 | 9 |
|---|---|---|---|---|---|---|---|---|---|
| Visibility Ranges | 50-200m | 200-500m | 500-1000m | 1-2 km | 2-4 km | 4-10 km | 10-20 km | 20-50 km | >50km |

































**Fig.1**. Map of the study area in Greece, including the Athens urban station (NOA) and a reference, non-urban station (HER) at Heraklion airport, Crete. The gray surface represents the boundary of the Greater Athens Area (GAA).

























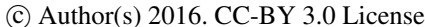


**Fig. 2a.** Main sectors related with air masses origin in Athens.



























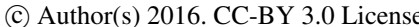

**Fig. 2b**. Seasonal variability of the relative frequency of air masses origin in Athens on the sectors defined in Fig. 2a, averaged over the period 2005-2014. Category 'L' refers to air masses of local origin.
















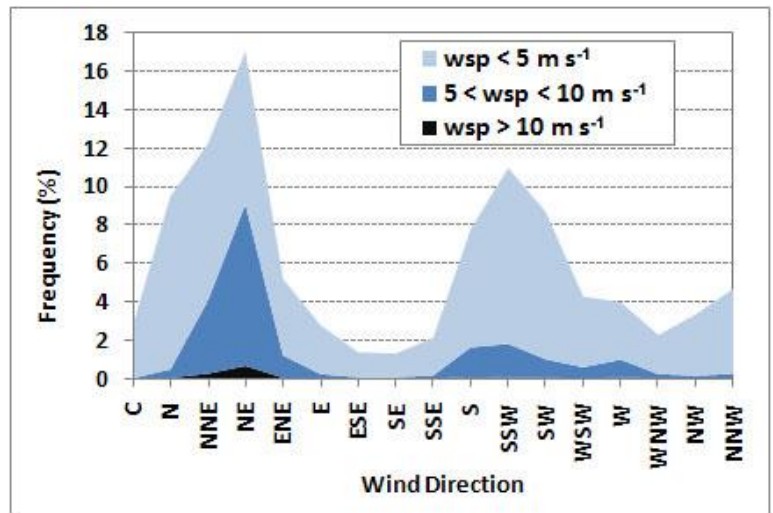


**Fig. 3**. Frequencies of surface wind directions for three wind speed (wsp) categories at NOA, based on hourly values of the period 1971-2000. For instance, the NE direction occurs cumulatively at a frequency of 17% which is the sum of 7.9 % (wsp < 5 m s$^{-1}$), 8.4 % (5 < wsp < 10 m s$^{-1}$) and 0.7 % (wsp > 10 m s$^{-1}$). The 'C' sector corresponds to calms (wsp < 0.3 m s$^{-1}$).















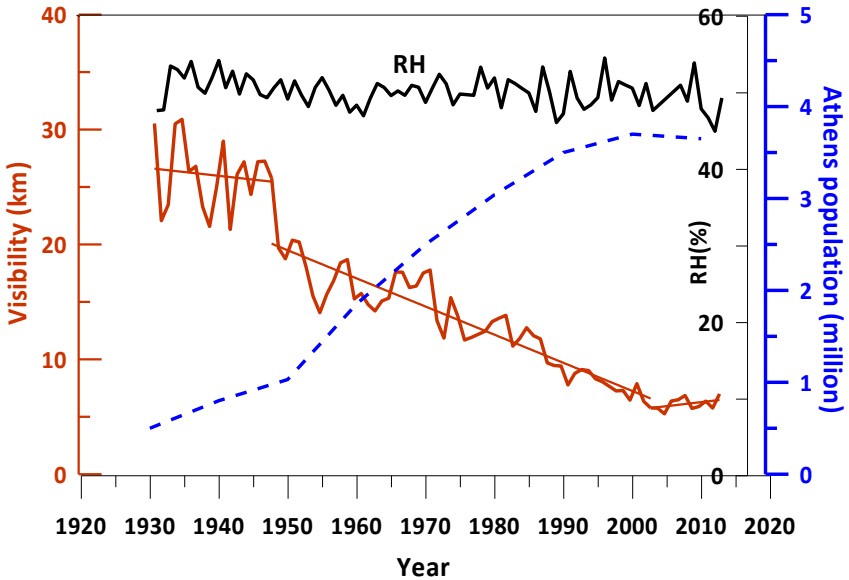


**Fig. 4**. Inter-decadal variability of the annual visibility in Athens from 1931 to 2013, along with linear trends for three sub-periods: 1931-1948, 1949-2003 and 2004-2013 (red line). The dashed blue line presents the population growth in Athens (in millions) since 1930 (Founda, 2011). The long-term variability of the annual relative humidity (RH) in Athens is also displayed (upper black line).






















**Fig. 5**. Frequency distribution of different visibility ranges (Table 2) in Athens for the three sub-periods, 1931-1948, 1949-2003 and 2004-2013.











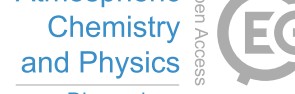





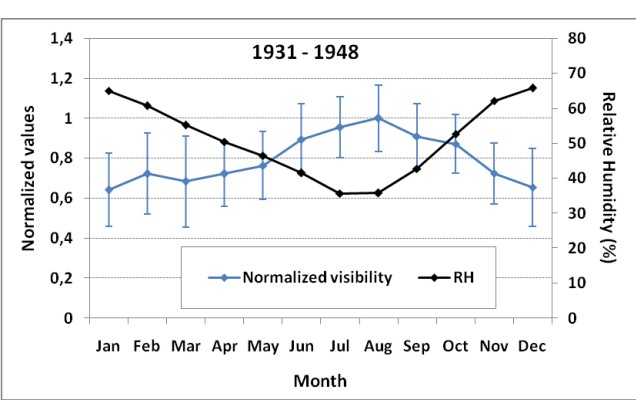


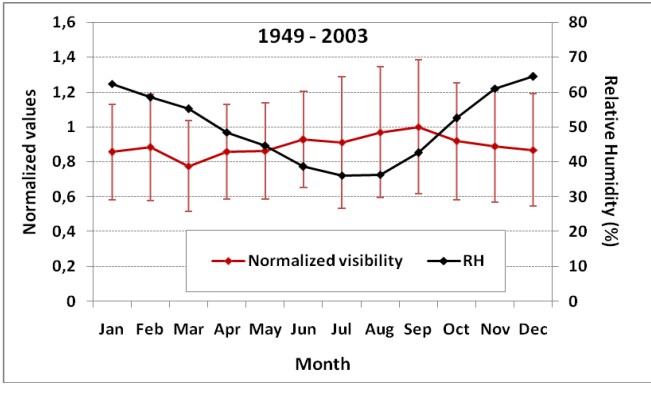


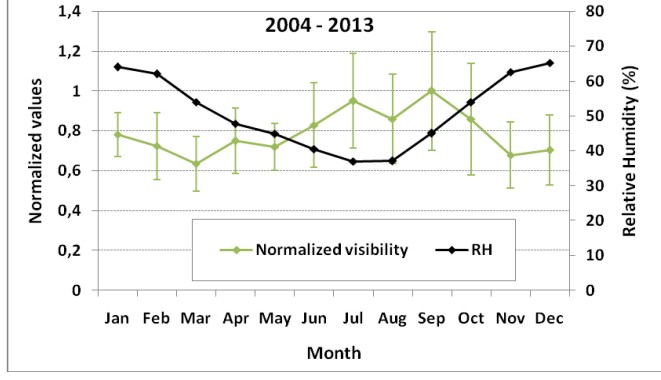


**Fig. 6**. Normalized mean monthly values of visibility in Athens for the three sub-periods, along with mean
monthly values of relative humidity (RH) for each sub-period. Vertical lines represent standard deviations of
mean monthly values of visibility.






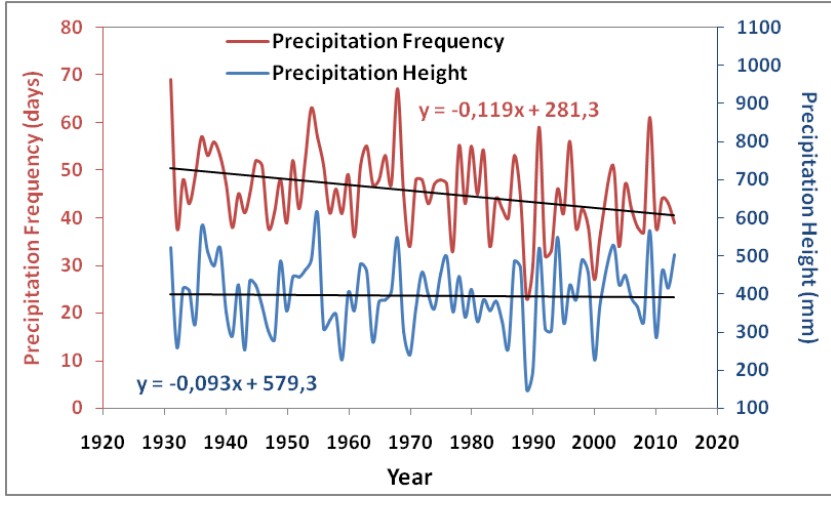


**Fig. 7**. Variation and long-term linear trends of the annual precipitation amount and frequency (number of days per year with precipitation > 1 mm) at NOA, over the period 1931-2013. Slopes of linear trends are also shown.











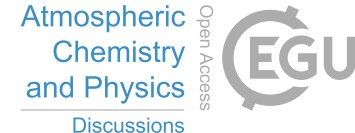












**Fig. 8**. Variation of visibility at NOA from 1931-2013 during the dry (May-Sep.), wet (Oct.-Mar.) and all year (Jan.-Dec.) period.





























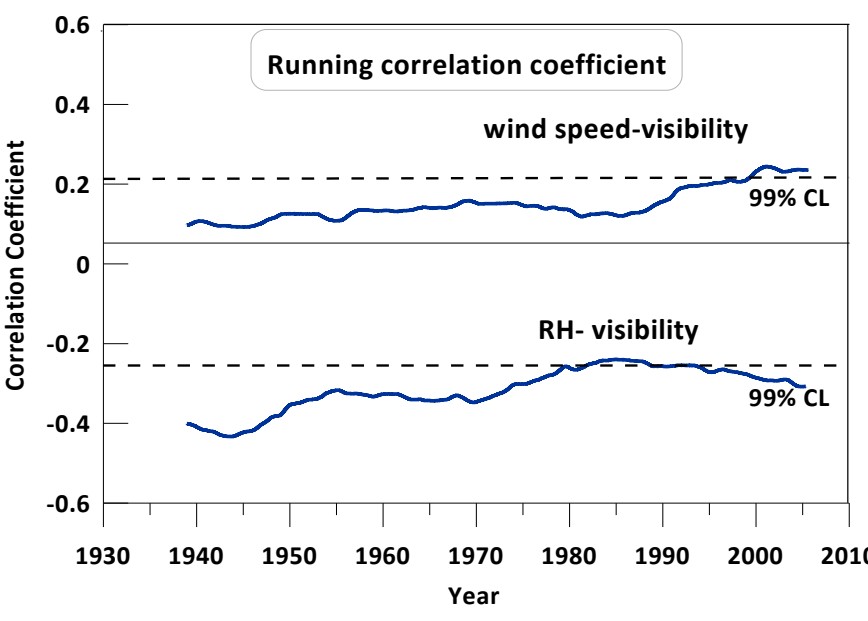

**Fig. 9**. Running correlation coefficient and confidence levels between visibility and wind speed (up) and visibility and RH (bottom) in Athens, over the period 1931-2013. A 15-yrs window was used.
























**Fig. 10**. Variation of visibility with wind direction (sectors) over the three sub-periods 1931-1948, 1949-2003 and 2004-2013. Visibility is normalized by its maximum value at a certain sector for each sub-period. Sector 'C' corresponds to calms (wind speed < 0.3 m s$^{-1}$). Frequency of each sector approximates closely its climatic value (Fig. 3) in all sub-periods.



















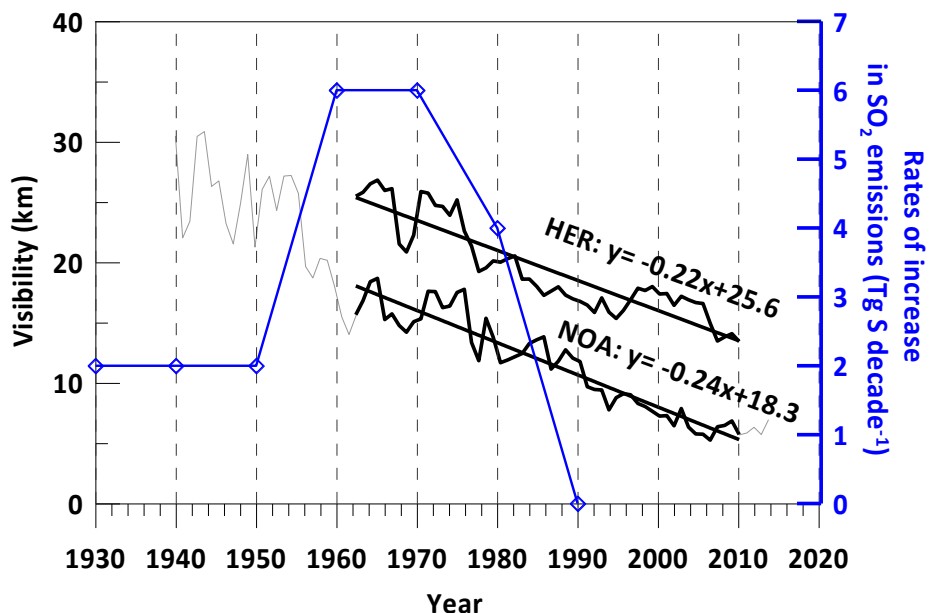


**Fig. 11**. Inter-decadal variability of the annual visibility at NOA (urban) and HER (background) stations. Bold black lines represent the common period of observations (1956-2009) at the two stations along with linear trends and slopes. Blue line illustates the rates of increase of $SO_2$ emissions in Europe (in Tg S decade$^{-1}$), as included in van Aardenne et al., 2001.
















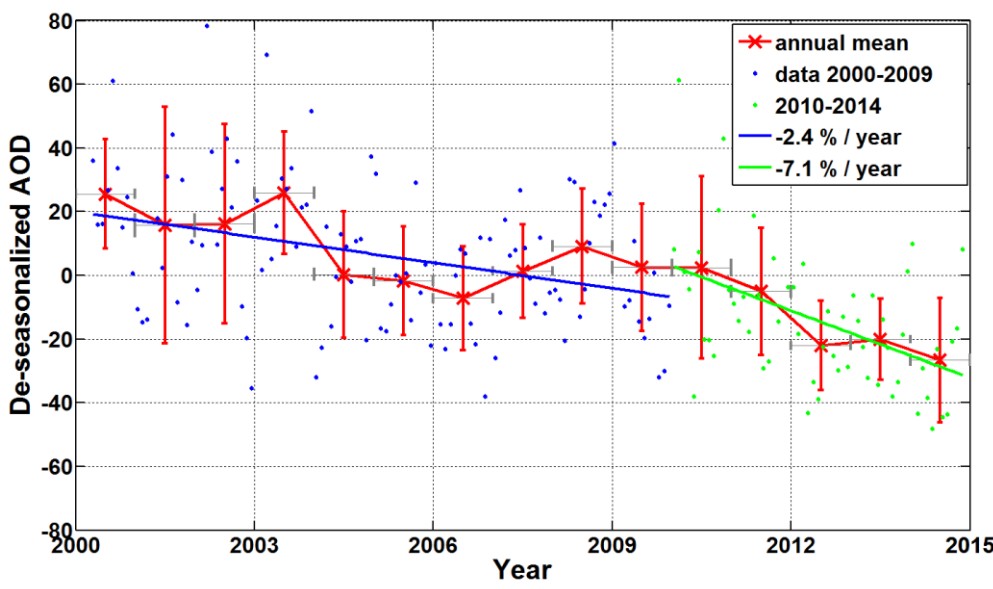


**Fig. 12**. Variability of deseasonalized monthly $AOD_{550nm}$ from 2000 to 2014 (red), along with linear trends for the
periods 2000-2009 (blue), 2010-2014 (green). Vertical bars describe the standard deviation of the annual value
based on the monthly ones and grey horizontal bars the respective year.











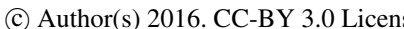





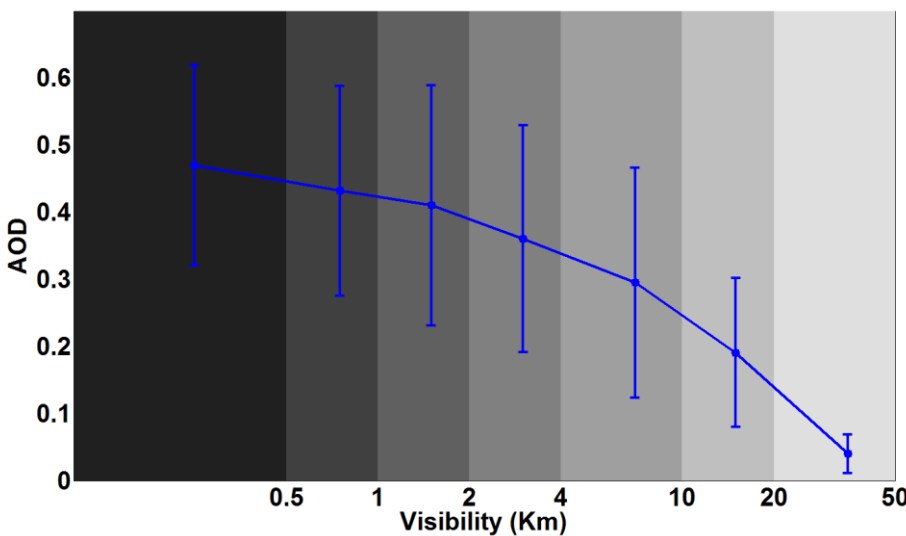


**Fig. 13**. MODIS AOD June-August mean values and standard deviations for each visibility index. Shaded areas represent visibility ranges (km) for each visibility class (Table 2) and points are plotted at the center of each visibility class.
















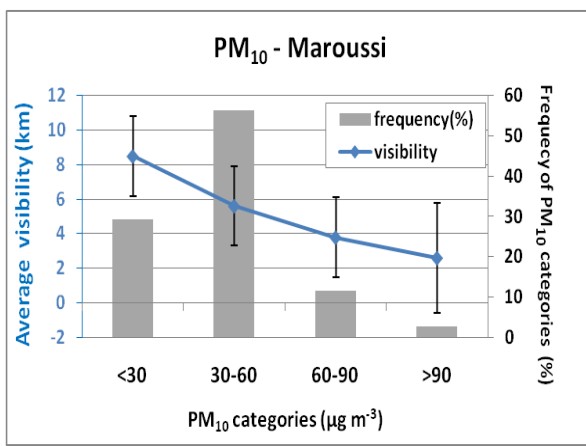
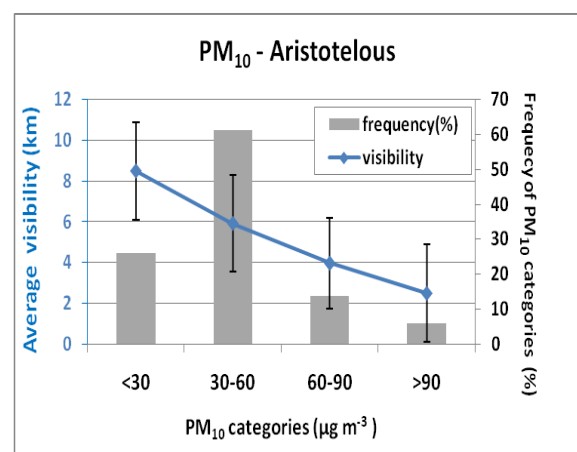


**Fig. 14**. Visibility as a function of different classes of $PM_{10}$ levels at an urban (Aristotelous) and a suburban (Maroussi) station in Athens. Measurements refer to the period 2008-2012. Geometric average and geometric standard deviations are applied on visibility observations. Frequencies of classes of $PM_{10}$ levels are also shown (grey bars).

























**Fig. 15**. Variation of the annual $PM_{10}$ concentrations at the reference station of Finokalia (Crete) over the period 2005-2014. Vertical lines represent standard deviations of the annual means.

