# Peer review of "Long-term visibility variation in Athens (1931-2013): A proxy for local and regional atmospheric aerosol loads (acp-2015-1025)"

_Atmospheric Chemistry and Physics, 2015_

## Referee Comment (RC1) · Anonymous Referee #2 · 28 Apr 2016

General: The study uses the long-time visibility records along with meteorological variables, emissions and satellite optical depth retrievals over Athens and explores the relationships between these variables over three distinctive sub-periods. The manuscript is clear, well-written with a very good introduction. However, I find the conclusions too long and can be substantially reduced by only pointing to the major outcomes of the study.

Minor comments

Line 237: How far from Athens? Characteristics of the site (emission sources etc)?

Lines 248-254: Better to present the trends in uniform units, per year in this case.

[Figure]

The resolutions of the excel-based figures should be improved.

The relation (zooming) between the two plots in Figure 1 is misleading.

Figure 7: Precipitation Height is misleading, drop the "Height"

Figure 13: Why the different bins have different widths? Does it stand for something? For instance why 0-0.5 km bin is largest? Please explain.

Figure 15: Can you also add the data for Athens here?

Technical corrections

Line 33: Remove the comma before (WMO, 1992).

Line 38: Replace "at" with "over"

Line 55: …. pollutant emissions….

Line 231: Correct as (Kanakidou et al., 2011)

Line 260 and 272: Correct "to 1 km" to "than 1 km"

Lines 261, 266 and 272: Correct "to 500 m" to "than 500 m"

Line 290: ..results IN improvement….

Line 345: Change "as regards" to "regarding"

Line 408: …. ARE due to local factors…

Line 423: ….in accordance WITH…

Line 475: INDEPENDENT of the location….

---

## Referee Comment (RC2) · Anonymous Referee #1 · 9 May 2016

In this article, Founda and coauthors describe the long term trend in visibility in Athens, Greece and compare this trend to meteorological variables, visibility changes at a non-urban site in the area, and satellite-derived aerosol optical depth values. The rapid degradation of visibility after 1950 and slight recovery since 2005 are correlated with meteorological conditions associated with air mass origin, PM10 surface measurements, and aerosol optical depth; these relationships suggest that visibility is a proxy for local and regional atmospheric aerosol levels. This trend and associated analyses provide a novel dataset for understanding long term changes in aerosol concentration near Athens. I'd suggest publication after the following comments have been addressed.

Major comments: 1) While the grouping of 3 periods of visibility trends are appropriate

when discussing changes over time, the middle period (1949-2003) is not appropriate when discussing frequencies (Figure 5) and seasonality (Figure 6) because the early part of the period has substantially different visibility conditions from the later period. When not showing a time series, the 1949-2003 period needs to be separated into several periods of more similar visibility conditions.

2) I think that a more comprehensive comparison between emissions changes and visibility trends would help improve the article. Figure 11 needs to have emissions on the y-axis as a magnitude rather than a rate of change, and plotting other types of emissions (NOx, EC, OC, etc) would be interesting to see if available. If the emission data could be segregated by air mass origin, it would be interesting to see if increases/decreases of emissions in certain parts of Europe have affected the visibility in Athens.

3) To add value in the visibility-satellite AOD comparison, I'd suggest examining the much longer-term dataset of AOD values from the Advanced Very High Resolution Radiometer (AVHRR) satellite. Although AVHRR retrieves AOD only over ocean grid cells, selecting the nearest ocean cell to Athens would enable an visibility-AOD comparison since 1981 when visibility values were still degrading.

Minor comment: 1) Many typos and text spacing problems persist in the document and have to be corrected. The first of many are listed by page number; line number (suggested correction): Page 1; Line 18 ("34%"), Page 1; Line 22 ("the 1950s"), Page 2; Line 46 ("containing"), Page 3; Line 82 ("oldest time"), Page 4; Line 90 ("construction"), Page 5; Line 118 ("...the year. The periods..."), Page 5; Line 129 ("Mediterranean"), Page 5; Line 136 ("60%"), Page 6; Line 173 ("Po Valley"), Page 7; Line 180 ("...subsequent reduction in vehicle use..."), Page 7; Line 201 ("with the naked eye."), Page 8; Line 208 ("Davis (1991)."), Page 10; Line 272 ("Overall, visibility did not exceed..."), Page 11; Line 312 ("different approaches, as for instance..."), Page 12; Line 343 ("...resulting in the reduction of visibility."), Page 13; Line 364 ("In other cases..."), Page 14; Line 491 ("increase of construction in the city."). I'd recommend an grammatical editor

to correct these and other errors prior to publication in final form.

2) Figure 2a should be referenced in the text before Figure 2b.

---

## Author Comment (AC1) · 21 Jun 2016

*__Author's response to referee#1__*

The authors are grateful to the anonymous referee # 1 for his/her time devoted to this
paper and the useful comments and suggestions, aiming at the improvement of the
manuscript.

All comments and recommendations of referee#1 were taken very seriously into
consideration for the preparation of a revised version of the manuscript. Additional effort
has been put to implement and incorporate suggestions in the manuscript to the best
possible degree and prepare a revised version accounting for all comments of referee #1.

In the following, we present our answers to the referee's comments as well as the changes
performed in the manuscript in the following order:

**A. Comments of referee #1**

**B. Authors' answers to each comment of referee #1**

**C. Changes in the manuscript to account for comments of referee #1**

*__A. Comments of  referee #1__*

*In this article, Founda and coauthors describe the long term trend in visibility in Athens,*
*Greece and compare this trend to meteorological variables, visibility changes at a nonurban*
*site in the area, and satellite-derived aerosol optical depth values. The rapid degradation of*
*visibility after 1950 and slight recovery since 2005 are correlated with meteorological*
*conditions associated with air mass origin, PM10 surface measurements, and aerosol optical*
*depth; these relationships suggest that visibility is a proxy for local and regional atmospheric*
*aerosol levels. This trend and associated analyses provide a novel dataset for understanding*
*long term changes in aerosol concentration near Athens. I'd suggest publication after the*
*following comments have been addressed.*
*__Major comments:__*

*1) While the grouping of 3 periods of visibility trends are appropriate when discussing*
*changes over time, the middle period (1949-2003) is not appropriate when discussing*
*frequencies (Figure 5) and seasonality (Figure 6) because the early part of the period has*
*substantially different visibility conditions from the later period. When not showing a time*
*series, the 1949-2003 period needs to be separated into*
*several periods of more similar visibility conditions.*

*2) I think that a more comprehensive comparison between emissions changes and*
*visibility trends would help improve the article. Figure 11 needs to have emissions*
*on the y-axis as a magnitude rather than a rate of change, and plotting other types*

*of emissions (NOx, EC, OC, etc) would be interesting to see if available. If the emission*

*data could be segregated by air mass origin, it would be interesting to see if*

*increases/decreases of emissions in certain parts of Europe have affected the visibility*

*in Athens.*

*3) To add value in the visibility-satellite AOD comparison, I'd suggest examining the much*

*longer-term dataset of AOD values from the Advanced Very High Resolution Radiometer*

*(AVHRR) satellite. Although AVHRR retrieves AOD only over ocean grid cells, selecting the*

*nearest ocean cell to Athens would enable an visibility-AOD comparison since 1981 when*

*visibility values were still degrading.*

***Minor comment:***

*1) Many typos and text spacing problems persist in the document and have to be corrected.*
*The first of many are listed by page number; line number (suggested correction): Page 1; Line*
*18 ("34%"), Page 1; Line 22 ("the 1950s"), Page 2; Line 46 ("containing"), Page 3; Line 82*
*("oldest time"), Page 4; Line 90 ("construction"), Page 5; Line 118 ("...the year. The*
*periods..."), Page 5; Line 129 ("Mediterranean"), Page 5; Line 136 ("60%"), Page 6; Line 173*
*("Po Valley"), Page 7; Line 180 ("...subsequent reduction in vehicle use..."), Page 7; Line 201*
*("with the naked eye."), Page 8; Line 208 ("Davis (1991)."), Page 10; Line 272 ("Overall,*
*visibility did not exceed..."), Page 11; Line 312 ("different approaches, as for instance..."),*
*Page 12; Line 343 ("...resulting in the reduction of visibility."), Page 13; Line 364 ("In other*
*cases..."), Page 14; Line 491 ("increase of construction in the city."). I'd recommend an*
*grammatical editor to correct these and other errors prior to publication in final form.*

*2) Figure 2a should be referenced in the text before Figure 2b.*

***B.  Author's answers to the comments of referee #1***

***Major Comments***

**1.** Indeed, the grouping of the historical time series was mainly indicated by the different
slopes of trends observed in the three sub-periods 1931-1948, 1949-2003 and 2004-2013.  It
is true that the early part of the much longer sub-period (1949-2003) is characterized by
different visibility conditions compared to the latter part. For this reason, the initial grouping
was maintained only in trend analysis. In all other cases, namely when studying frequency
distribution (Fig. 5), seasonality (Fig.6) but also variation of visibility with wind direction (Fig.
10), the long period 1949-2003, was further divided into two parts, 1949-1975 and 1976-

2003. Figures 5, 6 and 10 were reproduced, where the plots concerning the 1949-2003 sub-period, were replaced by plots for the periods 1949-1975 and 1976-2003 (see section C below). The text in the manuscript in sections 3.2, 3.3 and 4.4.2 was revised accordingly, accounting for the new information derived from this additional grouping.

**2.** Historical data of other types of emissions for Europe such as NOx and OC were also considered and discussed in the manuscript. Plot of rates of changes of $SO_2$ emissions in Fig. 11 of the manuscript was now replaced with the plot of $SO_2$ emissions as a magnitude for a more direct comparison with visibility variations. Moreover, a plot of historical NOx emissions for Europe was added in Fig. 11. Details are provided in section C below.

**3.** We have followed the reviewer's recommendation and used the AVHRR satellite data (available since 1981) in addition to support the current MODIS related analysis concerning AOD and visibility. The additional analysis was incorporated in the manuscript accordingly as described below (see section C).

**Minor comments of referee #1**

**1.** Although a grammatical editor had been already used, for some reason it didn't work properly and a number of grammatical and syntax errors remained in the text. Additional effort and a new editor have been used now to cope with this problem. All suggested first corrections by the referee were applied in the text. Additional syntax errors were also found and corrected.

**2.** This was corrected.

**C. Changes in the manuscript to account for the comments of referee#1**

1. In order to comply with major comment 1, Figs 5, 6 and 10 concerning the frequency distribution, seasonality and variation of visibility with wind direction were reproduced. In the new figures, the sub period 1949-2003 was replaced by two additional sub-periods, namely 1949-1975 and 1976-2003. The new Figures 5, 6 and 10 are displayed below

[Figure]

**New Fig. 5**. Frequency distribution of different visibility ranges (Table 2) in Athens for the sub-
periods, 1931-1948, 1949-1975, 1976-2003 and 2004-2013.

[Figure]

**New Fig. 6**. Normalized mean monthly values of visibility in Athens for the sub-periods 1931-1948,
1949-1975, 1976-2003 and 2004-2013, along with mean monthly values of relative humidity (RH) for
each sub-period. Vertical lines represent standard deviations of monthly visibility means.

[Figure]

**New Fig. 10**. Variation of visibility with wind direction (sectors) over the sub-periods 1931-1948, 1949-1975, 1976-2003 and 2004-2013. Visibility is normalized by its maximum value at a certain sector for each sub-period. Sector 'C' corresponds to calms (wind speed < 0.3 m s-1). Frequency of each sector approximates closely its climatic value (Fig. 3) in all sub-periods.

The text in the manuscript (sections 3.2, 3.3, 4.4.2 and Conclusions) was revised accordingly, to account for the new information derived from this additional grouping.

**2.** Historical data of other types of emissions in Europe such as NOx and OC were also considered and discussed in the analysis. Fig. 11 of the manuscript was reproduced. In the new figure, graph of the rates of changes of $SO_2$ emissions was replaced with graph of historical emissions as a magnitude, for a more direct comparison with visibility variations. Moreover, a plot of historical NOx emissions for Europe was added in Fig. 11. Historical emissions of $SO_2$ and NOx for Europe were now derived from the studies of Vestreng et al. (2007, Fig. 1) and Vestreng et al. (2008, Fig. 4 ) respectively since they provide updated emissions data.

[Figure]

**New Fig. 11**. Inter-decadal variability of the annual visibility at NOA (urban) and HER (background) stations. Bold black lines represent the common period of observations (1956-2009) at the two stations along with linear trends and slopes. Solid blue line illustrates historical European emissions of $SO_2$ as included in Vestreng et al., 2007 and blue dashed line illustrates historical European emissions of $NO_2$ as included in Vestreng et al., 2008.

Historical emissions and trends of Organic Carbon as included in the study of Bond et al., (2007, Fig. 6) were also discussed. The segregation of emissions according to air mass origin was also now discussed in the text. Information for segregation was based on the same studies but also the study of Mylona (1996) and van Aardenne et al., 2001. Emphasis is given on air masses from N, NE directions (North, Eastern Europe), since on an annual basis, air masses from the N and NE sectors dominate in the area of interest (Figs 2, 3 of the manuscript).

**3.** Section 2.5 has been changed including the reviewer's suggestion to include the AVHRR analysis in addition to the one of MODIS. In this section, we describe the data set used with the respective references and the specific analysis and data set details for the Athens case.

Then section 3.6 has been changed accordingly including the results of the analysis of AVHRR data. In addition, we have included a new figure (Fig. 12a) showing the AOD changes in Athens area from 1981 to 2009 based on AVHRR and we have superimposed the AVHRR related results to Figure 13, describing now the AOD - visibility index relationship from two different data sets.

[Figure]

**New Fig. 12**. a) Variability of deseasonalized monthly AVHRR-based $AOD_{630nm}$ from 1981 to 2009 (black), along with linear trends for the periods 1981-1997 (blue), 1998-2009 (green). Vertical bars describe the standard deviation of the annual value based on the monthly ones .b) Variability of MODIS-based deseasonalized monthly $AOD_{550nm}$ from 2000 to 2014 (red), along with linear trends for the periods 2000-2009 (blue), 2010-2014 (green). Vertical bars describe the standard deviation of the annual value based on the monthly ones and grey horizontal bars the respective year.

[Figure]

**New Fig. 13**. MODIS at 550nm (blue) (2000-2014) and AVHRR at 630nm (red) (1991-2009), AOD June-August mean values and standard deviations for each visibility index. Shaded areas represent visibility ranges (km) for each visibility class (Table 2). AOD averages have been represented here in the average distance from each class

**Minor comments of  referee #1**

**1.**  All suggested first corrections by the referee were applied in the text. Additional syntax errors were also found and corrected.

**2.**  Figure 2a is now referenced in the text before Fig. 2b.

---

## Author Comment (AC2) · 21 Jun 2016

*__Author's response to referee #2__*

The authors are grateful to the anonymous referee #2 for his/her time devoted to this paper and the useful comments and suggestions, aiming at the improvement of the manuscript.

All comments and recommendations of referee #2 were taken very seriously into consideration for the preparation of a revised version of the manuscript. Additional effort has been put in order to implement suggestions and incorporate them in the manuscript to the best possible degree and prepare a revised version accounting for all comments of referee.

In the following, we present our answers to the comments of referee #2 as well as the changes performed in the manuscript in the following order:

**A. Comments of referee #2**

**B. Authors' answers to each comment of referee #2**

**C. Changes in the manuscript to account for comments of referee #2**

*__A. Comments of referee #2__*

*__General:__*

*The study uses the long-time visibility records along with meteorological variables,emissions*

*and satellite optical depth retrievals over Athens and explores the relationships between*

*these variables over three distinctive sub-periods. The manuscript is clear, well-written with a*

*very good introduction. However, I find the conclusions too long and can be substantially*

*reduced by only pointing to the major outcomes of the study.*

*__Minor comments__*

*Line 237: How far from Athens? Characteristics of the site (emission sources etc)?*

*Lines 248-254: Better to present the trends in uniform units, per year in this case.*

*The resolutions of the excel-based figures should be improved.*

*The relation (zooming) between the two plots in Figure 1 is misleading.*

*Figure 7: Precipitation Height is misleading, drop the "Height"*

*Figure 13: Why the different bins have different widths? Does it stand for something?*

*For instance why 0-0.5 km bin is largest? Please explain.*

*Figure 15: Can you also add the data for Athens here?*

*__Technical corrections__*

*Line 33: Remove the comma before (WMO, 1992).*

*Line 38: Replace "at" with "over"*

*Line 55: : : :. pollutant emissions: : :.*

*Line 231: Correct as (Kanakidou et al., 2011)*

*Line 260 and 272: Correct "to 1 km" to "than 1 km"*

*Lines 261, 266 and 272: Correct "to 500 m" to "than 500 m"*

*Line 290: ..results IN improvement: : :.*

Line 345: Change "as regards" to "regarding"

Line 408: : : :. ARE due to local factors: : :

Line 423: : : :.in accordance WITH: : :

Line 475: INDEPENDENT of the location: : :.

**B. Author's answers to the comments of referee #2**

***General comments***

Section 4 summarizes the findings of the study but also discusses in detail
linkage/attribution between the main results of the analysis and possible causes. For this
reason, this section is long enough. However, the section was reduced in an effort to focus
on the main findings of the study and also avoid duplications.

***Minor comments***

Line 237: Additional information for the reference station of Finokalia (Crete) was included
in the text (see changes in the manuscript, below).

Lines 248-254: This was now corrected in the manuscript.

Some of the excel -based figures were reproduced using a different graphical tool. When not
possible, the resolution of excel- based figures was increased.

Indeed, the zooming between the two plots in Fig. 1 is not successful. Fig. 1 was recreated
(see below, changes in the manuscript).

Fig. 7: The figure was corrected

Fig. 13: The bin widths are based on the WMO definition on visibility class index. They are
not equal as visibility in km and WMO visibility index does not have a linear relationship. The
XX' axis is logarithmic.

Fig. 15. PM10 for two stations in Athens from 2004-2014 were added in the figure (see
below, changes in the manuscript)

**Technical corrections**

Although a grammatical editor had already been used, for some reason it didn't work properly and a number of grammatical and syntax errors remained in the text. Additional effort and a new editor have been used now to cope with this problem.  All suggested technical corrections were applied in the text.

*C. Changes in the manuscript to account for the comments of referee #2*

*General*

 The length of section 4 was reduced in the manuscript. The discussion focused on the main findings of the study and duplication of information or extended analyses were avoided.

*Minor comments*

 Information for Finokalia station is added in the manuscript:  The Finokalia station (35.240° N, 25.600° E) is located on the Northern coast of Crete, Greece, at a distance of approximately 320 Km to the south of Athens. There is no significant human activity within an area of approximately 15km around the station, mainly characterized by a scarce vegetation. The closest large urban area is the city of Heraklion (HER), (see map. of Fig. 1) with 150 000 inhabitants, and located 50 km West from Finokalia. Aerosols at the site are mainly transported from the Southern-Eastern  Europe and Northern Africa, and to a lesser extend from central and western Europe (Kouvarakis et al., 2000; Mihalopoulos et al.,1997).

Lines 248-254: The trends of visibility were now expressed as km $yr^{-1}$ in the manuscript.

Some of the excel -based figures were reproduced using a different graphical tool. When not possible, the resolution of excel- based figures was increased.

Fig. 1 was reproduced with a proper zooming.

[Figure]

New Fig.1.  Map of the study area in Greece, including the Athens urban station (NOA) and a
reference, non-urban station (HER) at Heraklion airport, Crete. The gray surface represents the
boundary of the Greater Athens Area (GAA).

Fig. 7: The word 'Height' was dropped.

Fig. 15: The figure was recreated including annual PM10 values for the two stations of

Maroussi and Aristotelous in Athens.

[Figure]

New Fig. 15. Variation of the annual PM10 concentrations at the reference station of Finokalia (Crete)
over the period  2005-2014 and at the stations of Maroussi and Aristotelous in Athens (2004-2014).

***Technical corrections***

Technical corrections suggested by referee #2 were applied in the manuscript (Lines 33, 38,

55, 231, 260, 272, 261, 266, 272, 290, 345, 408, 423, 475)

Additional syntax errors were also found and corrected.

---

## Editor Decision (ED1)

**Editor's decision about revised ms. acp-2015-1025 by D. Founda et al., entitled "Long term visibility variation in Athens (1931-2013): A proxy for local and regional atmospheric aerosol loads".**

*By François Dulac, 26 July 2016*

Thank you for your revision. I am glad to accept your paper for publication pending editorial and technical corrections listed hereafter. Please especially check many missing spaces between words. I confirm my suggestion of your paper to the ACP editorial board for a journal highlight.

Editorial and technical corrections:
-Line 24: insert a space before "visibility".
-Line 26: specify "AVHRR and MODIS satellite-derived"...
-Line 27: insert a space before "confirmed".
-Line 31: change the comma position in "daylight (WMO, 1992)".
-Line 34: "loading in".
-Line 45: insert a space before and after "play"¨.
-Line 57: insert a space before "A "; "decrease in".
-Line 75: in addition to within the abstract, specify again "aerosol optical depth (AOD)".
-Line 88: insert a space before "behind" and before "during".
-Line 91: insert a space after "load".
-Lines 109, 110, 115, 118, 132, 187, 220, 230, 283, 285, 292, 297 (within parentheses): insert a space between the number and its unit.
-Line 112: add a space before "wet"; change "for the eastern".
-Line 115: space before and not after ">".
-Line 118: no space before "August".
-Line 120: insert a space before "rainy".
-Line 126: specify "a 10-yr climatology (2005-2014)".
-Line 127: after "trajectories", specify the arrival level at which air trajectories are computed.
-Line 128: insert a space before "N Africa".
-Line 129: insert a space after "events" and after "Mediterranean".
-Line 130: "the Balkans".
-Line 133-135: better move the last sentence of this paragraph to its beginning (line 126); specify the type of trajectories computed regarding vertical movements.
-Line 141: replace "has" by "is".
-Line 166: insert a space after "Regarding".
-Line 174: insert a space after "AOD$_{550nm}$"
-Lines179: insert a space before "subsequent".
-Line 187: insert a space after "located".
-Line 190: insert "local standard time" after "14:00".
-Line 195: insert a space before "Meteorological".
-Line 215: insert a space after "origin".
-Lines 221-222: change to "2330-km wide swath" (missing hyphen and space).
-Line 227: remove the space after "±".
-Line 228: insert a space after "AOD".
-Line 234: replace "is" by "was".
-Line 240: insert a space after "daily".
-Line 241: insert a space before "The above".
-Lines 241- 242: change to "within the distance of 70 km from".

-Line 243: insert a space after "used".

-Line 244: insert a space before "It".

-Line 245: replace ":" by ";" between the two refs.

-Line 246: insert a space before "2011".

-Line 247: insert a space before and after "$PM_{10}$".

-Line 248": replace the comma by a space in "station,at".

-Line 255: shift the comma from before the opening parenthesis to after the closing one.

-Line 256: insert a space before "Africa".

-Line 260: "trends in".

-Line 261: insert a space after "development".

-You might wish to include a comment on Fig. 4: I guess that the year-to-year variability in the visibility plot of Fig. 4 is associated to the very different resolutions (in km) of the visibility classes at high and low visibility, resulting in a decreasing fluctuation with decreasing visibility.

-Line 266: insert a space before "-2.8" and before "p".

-Line 267: insert a space before "corresponding".

-Line 268: specify "statistical sensitivity tests".

-Line 270: "of the first".

-Line 272: remove the space after "<".

-Line 273: "insert a space before "recent".

-Line 276: insert a space after "3.2".

-Line 277: change to "three sub-periods as described above follows changing trends".

-Line 279: insert a space before "Figure".

-Line 280: specify "for those 4 different sub-periods".

-Line 282: insert a space before "between".

-Line 283: insert a space before "Very".

-Line 284: insert a space before "of" and before "poor".

-Line 285: insert a space before "than" and before "occurred".

-Lines 289, 292 and 294: insert a comma before " respectively".

-Line 290: insert a space before "The" and after "negligible".

-Line 292: better write "Visibility lower than"; add a space before "10".

-Line 297: insert a space before "rises".

-Line 298: insert a space after "lower".

-Lines 300 and 301: remove the space after ">".

-Line 303: add a comma before the year of the 2 references.

-Line 305: insert a space before "area".

-Line 316: insert a space before "improvement".

-Line 317: insert a space before "enhance".

-Line 319: insert a space after "cycle".

-Line 320: insert a space before "Although".

-Line 326: "value of 67%".

-Line 332: no space in "March-July".

-Line 348: "decrease in RH levels".

-Line 355: insert a space before "resulting".

-Line 356: insert a space before "precipitation".

-Line 357: insert a space before "at" ad after "trends".

-Line 355: insert a space before "As".

-Lines 360 and 362: "day decade$^{-1}$".

-Line 361: insert a space before "while".

-Line 363: insert a space before "<".

-Line 363: insert a space before "year".

-Line 370: insert a space after "regarding".

-Line 374: insert a space before "period".

-Line 378: insert a space before "other".

-Lines 380 and 393: replace "(p<0.01) by "at the 99% confidence level".

-Line 382: add "from the beginning".

-Line 383: from the figure, "until the mid-1990s" seems more appropriate than "untilthe late1970s".

-Line 385: insert a space after "influence".

-Line 409: insert a space before "higher".

-Line 418: ";" between the 2 references; "2007 and 2009".

-Line 421: "disposing of".

-Line 422: insert a space before "most".

-Line 423: insert a space after "year,".

-Line 427: insert a space within "the time".

-Line 428: insert a space before ">" and "<".

-Line 430: insert a space before "during".

-Line 433: remove the comma after "that".

-Lines 436, 438, 440: insert a space after "$SO_2$".

-Line 440: "increase in".

-Line 441: insert a space before "sharp".

-Lines 446 and 448: insert a space after "$NO_x$".

-Line 447: "tendency to".

-Line 448: "all emission sectors".

-Line 449: "by air mass".

-Line 450: "N-NE" (no space).

-Line 452: insert a space before "have".

-Line 455: insert a space before "According".

-Line 456-457: "total emissions in the 1970s".

-Line 459: insert a space before "Stjern"; change "et al. (2011)" by "et al., 2011)".

-Line 465: insert a space before "finding" and before "trends".

-Line 467: insert a space after "that".

-Lines 472 and 473: 3 spaces missing between words in each line.

-Line 474: insert a space before "unpublished".

-Line 475: Ref. by Vautard should come first due to alphabetic order and same year.

-Line 491: 2 spaces missing between words.

-Line 492: 4 spaces missing between words.

-Line 493: 2 spaces missing between words.

-Line 495: insert a space after "series".

-Line 497: insert a space after "2014".

-Line 507: remove the 1st occurrence of "km".

-Line 510-511: remove the sentence starting with "It has to be noted" since the information is already given in line 503.

-Line 511-512: better shift the last sentence of the paragraph after "Fig. 13" in line 502.

-Line 513: "consists".

-Line 514: "satellite-based".

-Line 521: insert a space after "Athens" and after "14".

-Lines 523, 526, 529, 531, 533, 537, and 578: insert a space after "$PM_{10}$".

-Line 524: insert a space before "Despite".

-Line 528: "decrease in".

-Line 529: insert a space before "between".

-Lines 530, 531 and 532 (2 times): remove the space after "<" or ">".

-Lines 531 and 532: insert a space before ">" or "<".

-Line 533: insert spaces in "of the annual".

-Line 535: "tendency in".

-Line 536: "scales".

-Line 537: "trend in"; insert a space after "slight".

-Line 541: "the long historical".

-Line 545: insert a space before "unique".

-Line 546: "into 4 sub-periods".

-Line 550: insert a space after "distinct".

-Line 554: insert a space before "Visibility".

-Line 555: remove "on visibility".

-Line 556: insert a space after "winds,".

-Line 560: replace "poorer" by "larger".

-Line 564: insert a space before and after "4 km"; change "corresponding to" by "observed during".

-Line 566: 4 spaces missing between words.

-Lines 573-575: to follow the order of figures, interchange the 2 sentences but leave "also" in the second.

-Line 577: insert a space before "consistent".

-Line 579: specify "new transport".

-Line 580: insert a space before "the".

-Line 582: insert a space before "Besides" and before "regional".

-Line 583": insert a space before "which".

-Line 584: "for the first time".

-Line 586: space missing witin "to the".

-Line 587: replace "as the" by "showing an".

-Line 588: specify "from the 1930s to the 2000s" at the end of the sentence.

-Line 590: insert a space after "Acknowledgements."; prefer English version with an additional "e".

-Line 594: 2 spaces missing between words.

-Lines 597-599: "The contributions" ... "are".

-Line 598: insert a space after "data".

-Line 616: no space in pages 649-660".

-Line 622: remove "vol. " and also "2013. " before doi.

-Lines 641, 699, 701, and 765: insert a space before the year of the reference.

-Lines 643, 654, 707, 714, 741, 767, at least: check missing spaces after commas.

-Line 648: "Renew.".

-Line 691: specify University of the given Thesis.

-Line 705: add a final dot.

-Line 716: remove the journal number after the volume number; you probably have a double space before "315".

-Table 1: homogenize the use of a space before and not after "±"; shift the space from before to after "1" in "(>1 mm)".

-Line 806: insert a space before "used".

-Table 2: insert a space between numbers and unit "m" or "km" (4 instances).

-Legends of Fig. 1, 2b, 3, 8, 9, and 13: insert a space between "Fig." and the number.-Line 837: specify "The grey surface in the coloured map represents".

-Line 879: replace "on the sectors" by "following the sectors".

-Fig. 3: "Relative frequency" is a more appropriate legend for the y axis; in the legend box within the plot, insert a space between numbers and units and between units and remove the space between numbers and character "<" or ">" (e.g. "5< wsp <10 m s$^{-1}$"); you should remove the small intermediate ticks on the horizontal axis since they do not correspond to anything.

-Lines 902-905: specify "Relative frequencies"; check many missing spaces; add a space between "wsp" and character "<" or ">"; add a space between units in "m s$^{-1}$" (4 instances); add a space within "10 m" in line 904; please specify that "100% is the integral of the upper curve." before "For instance" (as I understand).

-Line 937: "Relative frequency distribution".

Fig. 6: you can probably enlarge a bit the 4 plots and character size in legends for a better readability; you should use the range 10-80 (or 0-70) for the RH axis to match the 2 vertical axes on the same vertical grid lines.

-Line 953: insert a space before "Vertical".

-Fig. 7 and line 972: rather use "day yr$^{-1}$" than "days" or "days/year"; change the position of the space in ">1 mm".

-Fig. 9: Specify "15-yr running" in the title box of the plot; the upper "99% CL" should better be moved over the upper dashed line; I suggest using a different colour for the two curves, and write their title ("wind speed-visibility" or "RH-visibility") using the respective colour of the plot.

-Line 1020: write "and 99% confidence levels (CL; dashed lines) between".

-Fig. 10: use decimal dot and bold characters for numbers in the y axis legend.

-Line 1044: insert a space before and after "of".

-Lines 1066-1069: many spaces are missing after punctuation signs including ")", ".", and ","; specify "annual average visibility"; for the last reference, remove the comma and put the year within ().

-Fig. 12: to improve readability, please add intermediate ticks in the x-axis and use bolder points in plot 12a; possibly enlarge the width of the figure.

-Line 1082: insert a space before "b)".

-Line 1083: insert a space before "trends".

-Fig. 13: using green would be more appropriate than blue for the MODIS 550-nm wavelength.

-Line 1092: inset a space between numbers and unit "nm" (2 instances).

-Line 1093: insert a space before "mean".

-Line 1103: empty page?

-Fig. 14: use bold characters for numbers in y axes.

-Line 1110: missing space after "Frequencies" and "PM$_{10}$"; specify "Relative frequencies".

-Lines 1110 and 1131: missing space after "PM$_{10}$".

-Fig. 15: check quality of original, it looks a bit blurred.

-Supplement, top of p. 3: "This resulted into".

-Supplement, p. 4: "trends in visibility"; move the 1$^{st}$ instance of "days" after the right parenthesis; remove the space after "<" within the parentheses and at the end of the page.

-Figs. S3 and S4: please add intermediate ticks (every 2 or 5 yrs).